



**A FIELD VALIDATED SURROGATE MODEL FOR OPTIMUM PERFORMANCE OF IRRIGATED**
**CROPS IN REGIONS WITH SHALLOW SALTY GROUNDWATER**
Zhongyi Liu[1], Zailin Huo[1]*, Chaozi Wang[1], Limin Zhang[2], Xianghao Wang[1],
Guanhua Huang[1], Xu Xu[1], Tammo Siert Steenhuis[3]*
1. Center for Agricultural Water Research in China, China Agricultural University, Beijing, 100083,
PR China
2. School of Water Resources and Environment, China University of Geosciences, Beijing,
100083, PR, China
3. Department of Biological and Environmental Engineering, Cornell University, Ithaca, NY, USA.
Correspondence to: Zailin Huo (huozl@cau.edu.cn)
Tammo S. Steenhuis (tss1@cornell.edu)





## Abstract

Optimum performance of irrigated crops in regions with shallow saline groundwater
requires a careful balance between application of irrigation water and upward movement
of salinity from the groundwater. Few field validated surrogate models are available to
aid in the management of irrigation water under shallow groundwater conditions. The
objective of this research is to develop a model that can aid in the management using a
minimum of input data that is field validated. In this paper a 2-year field experiment was
carried out in the Hetao irrigation district in Inner Mongolia, China and a physically based
integrated surrogate model for arid irrigated areas with shallow groundwater was
developed and validated with the collected field data. The integrated model that links
crop growth with available water and salinity in the vadose zone is called Evaluation of
the Performance of Irrigated Crops and Soils (EPICS). EPICS recognizes that field capacity
is reached when the matric potential is equal to the height above the groundwater table
and thus not by a limiting hydraulic conductivity. In the field experiment, soil moisture
contents and soil salt conductivity at 5 depths in the top 100 cm, groundwater depth,
crop height, and leaf area index were measured in 2017 and 2018. The field results were
used for calibration and validation of EPICS. Simulated and observed data fitted
generally well during both calibration and validation. The EPICS model that can predict
crop growth, soil water, groundwater depth and soil salinity can aid in optimizing water
management in irrigation districts with shallow aquifers.
**Key words:** Surrogate hydrological model, irrigated crops, shallow aquifer



## 1. Introduction

Irrigation water is a scarce resource, especially in arid and semi-arid areas of the world.
Irrigation improves quality and quantity of food production; however, excess irrigation
and salinization remain one of the key challenges. Almost 20% of the irrigated land in
the world is affected by salinization and this percentage is still on the rise (Li et al., 2014).
Salinity affects agricultural production (Williams, 1999). Soil salinization and water
shortages, especially associated with surface irrigated agriculture in arid to semi-arid
areas, is a threat to the well-being of local communities in these areas (Dehaan and
Taylor, 2002; Rengasamy, 2006).
In arid and semi-arid surface irrigation districts without a drainage infrastructure, the
groundwater table is close to the surface because more water has been applied than
crop evapotranspiration. Capillary rise of the shallow groundwater can be used to
supplement irrigation and thereby, in closed basins, can possibly save water for irrigating
additional areas downstream (Gao et al., 2015; Yeh and Famiglietti, 2009; Luo and
Sophocleous, 2010.). However, at the same time, capillary upward moving water carries
salt from the groundwater increasing the salt in the upper layers of the soil leading to
soil degradation and possibly decreasing yields and change of crop patterns to more salt
tolerant crops (Guo et al., 2018; Huang et al., 2018). Over 50% of the total irrigated
cropland, 5250 km$^2$ in the Hetao irrigation district in the Yellow River basin, is affected
by salinity (Feng et al., 2005). Therefore, understanding the interaction of improved crop
yield, soil salinization and decreased surface irrigation is important to the sustainability
of the surface irrigation water systems in arid and semi-arid areas. This will require



experimentation under realistic farmers' field conditions, as well as modeling to extend
the findings beyond the plot scale.

Field scale models for water, solute transport and crop growth are widely available.

Crop growth models use either empirical functions or model the underlying physiological
processes (Liu, 2009). Models widely used for simulating crop growth are EPIC (Williams
et al., 1989), DSSAT (Uehara, 1989), WOFOST (Diepen et al., 1989) and AquaCrop
(Hsiao et al., 2009; Raes et al., 2009; Steduto et al., 2009). Models focused on water
and solute movement in the vadose zone using some form of Richards' equation are
HYDRUS (Šimůnek et al., 1998) and SWAP (Dam et al., 1997). Models that integrate crop
growth and water-solute movement processes are SWAP-WOFOST (Hu et al., 2019),
SWAP-EPIC (Xu et al., 2015; Xu et al., 2016), HYDRUS-EPIC ((Wang et al., 2015), and
HYDRUS-DSSAT (Shelia et al., 2018). These integrated models require input data that are
usually not available when applied over extended areas (Liu et al., 2009; Xu et al., 2016;
Hu et al., 2019). The EPIC crop growth model is often preferred in integrated crop
growth hydrology models because it requires relatively few input data and is accurate
(Wang et al., 2014; Xu et al., 2013; Chen et al., 2019).

There is a tendency with the advancement of computer technology to include more

physical processes in these models (Asher et al., 2015; Doherty and Simmons, 2013;
Leube et al., 2012). Detailed spatially input of soil hydrological properties and crop
growth are required to take advantage of the model complexity (Flint et al., 2002; Rosa
et al., 2012). This greater model complexity, both in space and time, requires longer
model run times, especially for the time-dependent models (Leube et al., 2012). These



models are useful for research purposes but for actual field applications, the required
input data are not available and expensive to obtain. In such cases, simpler surrogate
models are a good alternative (Blanning, 1975; Willcox and Peraire, 2002; Regis and
Shoemaker, 2005). Surrogate models run faster and are as accurate as the complex
models for a specific problem (shallow groundwater here) but not as versatile as the
more complex models that can be applied over a wide range of conditions (Asher et al.,

2015).

Simple surrogate models are abundant in China for areas where the groundwater is

deeper than approximately 10 m (Kendy et al., 2003; Chen et al., 2010; Ma et al., 2013;
Li et al., 2017), but are limited and relatively scarce for areas where the goundwater is
near the surface in the arid to semi-arid areas (Xue et al., 2018; Gao et al., 2017; Liu et
al., 2019). When the groundwater is deep, the change in matric potential in the subsoil is
small and the hydraulic potential is equal to the gravity potential. However, for areas with
shallow aquifers (i.e., less than approximately 3 m), the matric potential cannot be
ignored. The flow of water is upward when the absolute value of matric potential is
greater than the groundwater depth or downward when it is less than the groundwater
depth (Gardner, 1958; Gardner et al., 1970a; b; Steenhuis et al., 1988). The field
capacity in these soils is reached when the hydraulic gradient is constant (i.e., the
constant value of sum of matric potential and gravity potential). In this case, the soil
water is in equilibrium and no flow occurs.

Because of the shortcomings in the above complex models, the objective of this

research was to develop a field validated surrogate model that could be used to optimize



both water use efficiency and crop yield in irrigated areas with shallow groundwater and
salinized soil with a minimum of input parameters. To validate the surrogate model, we
performed a 2-year field experiment in the Hetao irrigation district that investigated the
change in soil salinity, moisture content, groundwater depth and maize and sunflower
growth during the growing season.

**2. Model description**
2.1 Introduction of the model
In a recent study, we presented a surrogate model for the vadose zone with shallow
groundwater using the novel concept that the moisture content at field capacity is a
unique function of the groundwater depth after irrigation or precipitation that wets up
the entire soil profile. The model, called the Shallow Vadose Groundwater model, applies
directly to surface irrigated districts where the groundwater is within 3.3 m from the soil
surface (Liu et al. 2019). The model was a proof of concept with calibrated values for
evapotranspiration and soil salinity and was not simulated.
To make the Shallow Vadose Groundwater model more physically realistic, we added
a crop growth model and included the effect of salinity and moisture content on
evaporation and transpiration directly in this study. The new model that combines parts
of the Environmental Policy Integrated Climate (EPIC) with Shallow Vadose Groundwater
model is called the *Evaluation of the Performance of Irrigated Crops and Soils* (EPICS).
2.2 Structure of the EPICS model
In the EPICS model, the soil profile is divided into five layers of 20 cm (from the soil





surface down) and a sixth layer that stretches from the 100 cm depth to the water table
below (Fig. 1).

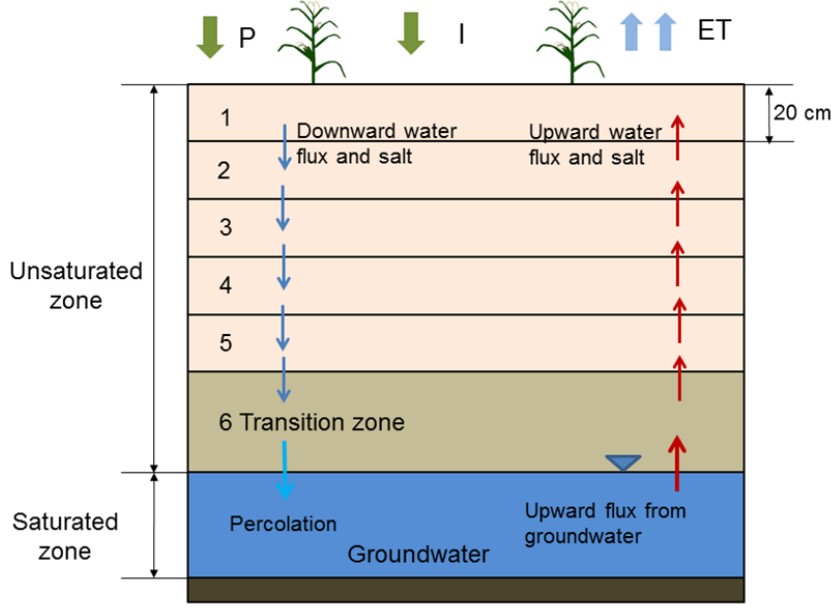


Fig 1. Schematic diagram of model components and water movement

The moisture content and salt content are calculated for each day (Fig.1). All flow

takes place within the day and the water and salt content are in "equilibrium" (i.e., fluxes
are zero) at the end of the day for which the calculations are made. Daily fluxes
considered in the vadose model are the following: at the surface, the fluxes are irrigation,
both irrigation water, $I(t)$, and salt, $S_0(t)$, and precipitation, $P(t)$, and for each layer, $j$, on
days with irrigation and rainfall, the downward flux of water, $R_w(j,t)$, and salt, $S(j,t)$,
between the layers. On days without water input at the soil surface, an upward
groundwater flux $U(j,h,t)$, and salt, $S(j,t)$ are considered. The flux to the surface depends
on the groundwater depth. Finally, transpiration, $T(j,t)$, removes water from the layers
with roots of the crops and evaporation, $E(j,t)$, from all layers.





The EPICS model consists of two modules: the VADOSE module and the CROP
module. The two modules are linked through the evapotranspiration flux in the soil (Fig.

2).

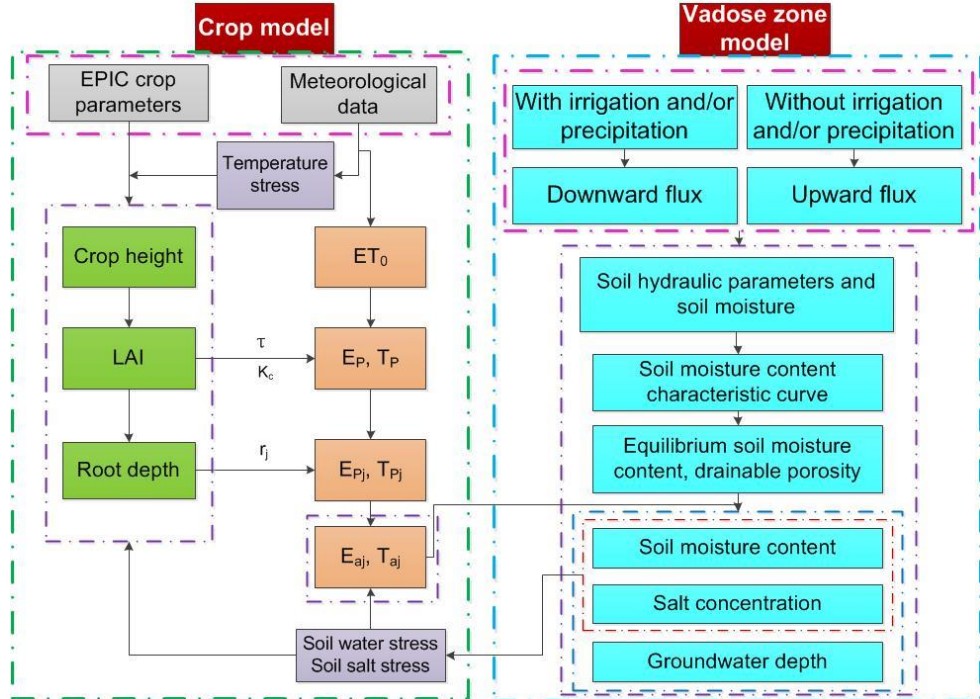


Fig 2. Schematic diagram of the linked novel Shallow Aquifer-Vadose zone surrogate
module and EPIC module. Note: $ET_O$ is the reference evapotranspiration, $E_p$ and $T_p$ are the
potential evaporation and potential transpiration, $E_a$ and $T_a$ are the actual evaporation
and actual transpiration, $K_c$ is the crop coefficient, $\tau$ is the development stage of the leaf
canopy, and $r_j$ is the root function of soil layer $j$.

The CROP module employs functions of the EPIC model (Williams et al., 1989) and
root growth distribution (Novak, 1987; Kendy et al., 2003; Chen et al. 2019). The CROP
module calculates daily values of crop height, root depth and leaf area index (LAI) based
on climatic data (Fig. 2).
The VADOSE module calculates the moisture and salt content in the root zone and



the upward movement of the groundwater (Fig.2). Field capacity varies with depth and is
a function of the (shallow) groundwater depth and the soil characteristic curve (Liu et al.,
2019). Moisture contents become less than field capacity when the upward flux is less
than the actual evapotranspiration.

Finally, the link between the VADOSE and the CROP modules is achieved by

calculating the actual evapotranspiration with parameters of both modules consisting of
the moisture content and the salt content simulated in the VADOSE module and root
distribution and potential evapotranspiration in the CROP module (Fig. 2).
2.3 Theoretical background of the EPICS model
In the next section, the equations of the CROP in the VADOSE modules are presented.
The calculations are carried out sequentially on a daily time step. Finer resolution is not
needed for managing water and salt content for irrigation. In the first step, the actual
evaporation and transpiration are calculated for each layer in the model. Next, the
moisture content and salt content are adjusted for the various fluxes. Since the equations
for the downward movement on days of rainfall and/or irrigation are different than for
upward movement from the groundwater on the remaining days, we present upward and
downward movement in separate sections. The code was written in Matlab 2014a and
Microsoft Excel was used for data input and output.
*2.3.1 CROP module*
The crop module uses functions of EPIC (Erosion Productivity Impact Calculator, Williams
et al., 1989) to calculate leaf area index, LAI, crop height and the root depth (green





boxes in Fig. 2), and the potential transpiration, $T$, and evaporation, $E$ (orange boxes in
Fig, 2). Input data for the CROP module included: mean daily temperature ($T_{mean}$),
maximum daily temperature ($T_{mx}$), minimum daily temperature ($T_{mn}$), maximum crop height
($H_{mx}$), maximum LAI ($LAI_{mx}$), maximum root depth ($RD_{mx}$), dimensionless canopy extinction
coefficient ($K_b$), and total potential heat units required for crop maturation ($PHU$).

The potential rates of evaporation, $E_P(j,t)$, and transpiration, $T_P(j,t)$, of different

layers are derived from the total rates and a root function that determines the
distribution of roots in the vadose zone

$$T_P(j,t) = r_T(j,t)T_p(t) \qquad (1a)$$

$$E_p(j,t) = r_E(j,t)E_p(t) \qquad (1b)$$

where the letters in the parenthesis are the independent variables on which the
parameter before the parenthesis depends, $T_P(t)$ is the total potential transpiration and
$E_P(t)$ is the total potential transpiration at time, $t$. Both are calculated with the CROP
module (S1 in the supplementary material). Root functions (Sau et al., 2004; Delonge et
al., 2012) were used to calculate transpiration and evaporation of different soil layer.
$r_T(j,t)$ is the root function for the transpiration and $r_E(j,t)$ is the root function for the
evaporation. Both have the same general equation but with a different value for the
constant $\delta$.

$$r_T(j,t) = \left[\frac{1}{1 - exp(-\delta)}\right]\left\{exp\left[-\delta\left(\frac{Z_{1j}}{Z_{2j}}\right)\right]\left[1 - exp\left(-\delta\frac{Z_{2j} - Z_{1j}}{Z_r}\right)\right]\right\} \qquad (2a)$$

$$r_E(j,t) = \left[\frac{1}{1 - exp(-\delta)}\right]\left\{exp\left[-\delta\left(\frac{Z_{1j}}{Z_{2j}}\right)\right]\left[1 - exp\left(-\delta\frac{Z_{2j} - Z_{1j}}{Z_r}\right)\right]\right\} \qquad (2b)$$

Where $z_{1j}$ is the depth of the upper boundaries of the soil layer $j$. For $r_T(j,t)$ if the root




depth is smaller than the lower boundaries of the soil layer $j$, $Z_{2j}$ is equal to the root
depth and if the root depth is greater than the lower boundaries of the soil layer $j$, $Z_{2j}$ is
the depth of the lower boundaries of the soil layer $j$. For $r_E(j,t)$, $Z_{2j}$ is depth of the
lower boundaries of the soil layer $j$. $Z_r$ is the root zone depth and $\delta$ is the water use
distribution parameter. Note that the sum of $r_T(j,t)$ of all soil layers is equal to 1. In the
study of Novark (1987), the value of $\delta$ for corn is 3.64 and we used this value. To obtain
$r_E(j,t)$, $\delta$ was set to 10 (Chen et al., 2019; Kendy et al., 2003). Sunflower root function
simulation employed the same $\delta$ values as for maize.

The actual evaporation rates, $E_a(j,t)$, and transpiration, $T_a(j,t)$, for each soil layer, $j$,

at time, $t$, are calculated as a proportion of the potential values as:
$$E_a(j,t) = k_E(j,t)E_p(j,t) \tag{3a}$$

$$T_a(j,t) = k_T(j,t)S(j,t)T_p(j,t) \tag{3b}$$

where $k_E(j)$ and $k_T(j)$ are water stress coefficients and $S(j)$ is a salt stress coefficient.
According to Raes et al. (2009), the water stress coefficients are
$$k_E(j,t) = \exp\left(-2.5\frac{\theta_{0.33}(j)-\theta(j,t)}{\theta_{0.33}(j)-\theta_{15}(j)}\right) \qquad \theta \leq \theta_{0.33} \tag{4a}$$

$$k_E(j,t) = 1 \qquad \theta > \theta_{0.33} \tag{4b}$$

where $\theta_{0.33}(j)$ is the moisture content at 0.33 bar or -33 kPa for layer $j$, or when the
conductivity becomes limiting and $\theta_{15}(j)$ is the moisture content at wilting point 15 bar
(1.5 Mpa), $\theta(j,t)$ is the soil moisture content for layer $j$ at time $t$.
Then water stress coefficient in Eq. 3b is:
$$k_T(j,t) = 1 - \frac{\exp\left[\left(1-\frac{\theta(j,t)-\theta_{15}(j)}{(1-p)[\theta_{0.33}(j)-\theta_{15}(j)]}\right)f_{shape}\right]-1}{\exp(f_{shape})-1} \qquad \theta \leq \theta_{0.33} \tag{5a}$$

$$k_T(j,t) = 1 \qquad \theta > \theta_{0.33} \tag{5b}$$



where $f_{shape}$ is the shape factor of $k_T(j,t)$ curve, $\rho$ is the fraction of readily available
soil water relative to the total available soil water. Finally, the salt stress coefficient
$S(j,t)$ for each layer in Eq 3b can be calculated as (Allen et al., 1998; Xue et al., 2018):

$$S(j,t) = 1 - \frac{B}{100\,k_y}(EC_e(j,t) - EC_{ethreshold}) \qquad (6)$$

where $k_y$ is the factor that affects the yield, $EC_e$ is the electrical conductivity of the soil
saturation extract (ms cm⁻¹), $EC_{ethreshold}$ is the calibrated threshold of the electrical
conductivity of the soil saturation extract when the crop yield becomes affected by salt
(ms cm⁻¹), and $B$ is the calibrated crop specific parameter that describes the decrease rate
of crop yield when $EC_e$ increases per unit below the threshold. The electrical
conductivity of the soil saturation extract can be calculated as (Rhoades et al., 1989):

$$EC_e = 1.33 + 5.88 \times EC_{1:5} \qquad (7)$$

where $EC_{1:5}$ is the electrical conductivity of the soil extract that soil samples mixed with
distilled water in a proportion of 1:5.
*2.3.2 VADOSE Module*
2.3.2.1 Moisture content at field capacity
Field capacity with a shallow groundwater is different than in soils with deep
groundwater where water stops moving when the hydraulic conductivity becomes
limiting at -33 kPa. When the groundwater is shallow, the hydraulic conductivity is not
limiting and the water stops moving when the hydraulic potential is constant and thus
the matric potential is equal to the height above the water table (Gardner 1958; Gardner
et al.,1970a, b; Steenhuis et al. 1988; Liu et al., 2019). Assuming a unique relationship
between moisture content and matric potential (i.e. soil characteristic curve), the moisture

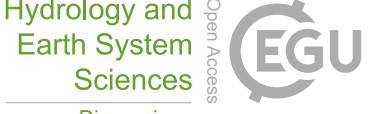

content at any point above the water table is a unique function of the water table depth.
Thus, any water added above field capacity will drain downward. When the groundwater
is recharged, the water table will rise and increase the moisture contents at field capacity
throughout the profile.

The moisture contents at field capacity were found by Liu et al. (2019) using the

simplified Brooks and Corey soil characteristic curve (Brooks and Corey, 1964)
$$\theta = \theta_s \left[\frac{\varphi_m}{\varphi_b}\right]^{-\lambda} \qquad for \ |\varphi_m| > |\varphi_b| \qquad (8a)$$

$$\theta = \theta_s \qquad for \ |\varphi_m| \leq |\varphi_b| \qquad (8b)$$

in which $\theta$ is the soil moisture content (cm$^3$ cm-$^3$), $\theta_s$ is the saturated moisture content
(cm$^3$ cm-$^3$), $\varphi_b$ is the bubbling pressure (cm), $\varphi_m$ is matric potential (cm), and $\lambda$ is the
pore size distribution index. The moisture content at field capacity, $\theta_{fc}(z, h)$, for any
point, z, from the surface water for a groundwater at depth, h, can be expressed as (Liu et
al. 2019)
$$\theta_{fc}(z, h) = \theta_s(z) \left[\frac{h - z}{\varphi_b}\right]^{-\lambda} \qquad for \ |h - z| > |\varphi_b(z)| \quad (9a)$$

$$\theta_{fc}(z, h) = \theta_s(z) \qquad for \ |h - z| \leq |\varphi_b(z)| \quad (9b)$$

where *h* is the depth of the groundwater and *z* (cm) is the depth of the point below the
soil surface. Thus (*h-z)* is the height above the groundwater and this is equal to the
matric potential for soil moisture content at field capacity.

For shallow groundwater, the matric potential at the surface is -33kPa when the

groundwater is 3.3 m depth. For this matric potential, as mentioned above, the
conductivity becomes limiting. This depth of the groundwater is therefore the lower limit
over which the VADOSE module is valid.


Evapotranspiration can lower the soil moisture content below field capacity. Thus,
the maximum moisture content in the VADOSE module is determined by the soil
characteristic curve and the height of the groundwater table, and the minimum is the
wilting point that can be obtained by evapotranspiration by the crop. Note that the
saturated hydraulic conductivity does not play a role in determining the moisture content
because inherently it is assumed that it is not limiting in the distribution of the water.
2.3.2.2 Drainable porosity
The drainable porosity that is a function of the depth is calculated first because it is
independent of time. The drainable porosity is obtained by calculating the field capacity,
$W_{fc}(h)$ (cm) for each layer at all groundwater depths. The total water content at field
capacity of the soil profile over a prescribed depth with a water table at depth $h$ can be
expressed as:

$$W_{fc}(h) = \sum_{j=1}^{n} \left[ L(j)\, \theta_{fc}(j,h) \right] \qquad (10)$$

where $\theta_{fc}(j,h)$ is the average moisture content at field capacity of layer j that can be
found by integrating Eq. 8 from the upper to the lower boundary of the layer and
dividing by the length $L(j)$ which is the height of layer $j$. The matric potential at the
boundary is equal to the height above the water table. The drainable porosity, $\mu(h)$,
which is a function of the groundwater depth $h$, can simply be found as the difference in
water content when the water table is lowered over a distance of $2\Delta h$.

$$\mu(h) = \frac{W_{fc}(h + \Delta h) - W_{fc}(h - \Delta h)}{2\Delta h} \qquad (11)$$

where $\Delta h = 0.5L(j)$.



2.3.2.3 Downward flux (at times of irrigation and/or precipitation)
**Water**
A downward flux occurs when either the precipitation or irrigation is greater than the
actual evapotranspiration. In this case, upward flux will not occur because the actual
evapotranspiration is subtracted from the input at the surface. We consider two cases
when the groundwater is being recharged and when it is not.

When the net flux at the surface (irrigation plus rainfall minus actual

evapotranspiration) is greater than that needed to bring the soil up to equilibrium
moisture content, the groundwater will be recharged and the distance of the
groundwater to soil surface decreases and the moisture content will be equal to the
moisture at field capacity. The fluxes from one layer to the next can be calculated simply
by summing the amount of water needed to fill up each layer below to the new moisture
content at field capacity. Hence, the percolation to groundwater, $R_{gw}(t)$, can be
expressed as:
$$R_{gw}(t) = P(t) + I(t) - E_a(t) - T_a(t) - \sum_{j=1}^{n} \frac{\left[\theta_{fc}(j,h) - \theta(j,t-\Delta t)\right]L(j)}{\Delta t} \qquad (12)$$

where $n$ is the total number of layers, $\theta(j,t)$ is the average soil moisture content in day
$t$ of layer $j$, $E_a(t)$ is the actual evaporation, $T_a(t)$ is the actual transpiration, $P(t)$ is
the precipitation, and $I(t)$ is the irrigation.

When the groundwater is not recharged, the rainfall and the irrigation are added to

uppermost soil layer and when the soil moisture content will be brought up to the field
capacity and the excess water will infiltrate to next soil layer bringing it up to field



capacity. This process continues until all the rainwater is distributed. Formally the soil
moisture can be expressed as

$$\theta(j,t) = min\left[\theta_{fc}(j,h), \left[\theta(j,t-\Delta t) + \frac{R_w(j-1,t)\,\Delta t}{L(j)}\right]\right] \quad (13)$$

where $\theta(j,t)$ is the average soil moisture content in day $t$ of layer $j$, $R_w(j-1,t)$ is the
percolation rate to layer $j$ and can be found with Eq 12 by replacing j-1 for n in the
summation sign.

$$R_w(j-1,t) = P(t) + I(t) - E_a(t) - T_a(t) - \sum_{1}^{j-1}\frac{[\theta_{fc}(j,h)-\theta(j,t-\Delta t)]L(j)}{\Delta t} \quad (14)$$

For the uppermost soil layer, the water percolation can be expressed as

$$R_w(0,t) = I(t) + P(t) - E_a(t) - T_a(t) \quad (15)$$

**Salinity**
The salt concentration for layer j can be expressed by a simple mass balance as:

$$C(j,t) = \frac{\theta(j,t-\Delta t)\,C(j,t-\Delta t)L(j) + [R_w(j-1,t)\,C(j-1,t) - R_w(j,t)\,C(j,t)]\,\Delta t}{\theta(j,t)L(j)} \quad (16)$$

where $C(j,t)$ is the salt concentration of layer j at time t (g L⁻¹). The equation can be
rewritten as an explicit function of $C(j,t)$

$$C(j,t) = \left[\frac{\theta(j,t)L(j)}{1 + R_w(j,t)\,\Delta t}\right]\left[\frac{\theta(j,t-\Delta t)\,C(j,t-\Delta t)L(j) + R_w(j-1,t)\,C(j-1,t)\,\Delta t}{\theta(j,t)L(j)}\right] \quad (17)$$

For the surface layer j=1, we obtain

$$C(1,t) = \left[\frac{\theta(1,t)L(1)}{1 + R_w(1,t)\Delta t}\right]\left[\frac{\theta(1,t)L(1)}{1 + R_w(1,t)\Delta t}\frac{\theta(j,t-\Delta t)\,C(j,t-\Delta t)L(j) + I(t)\,C_I\,\Delta t}{\theta(j,t)L(j)}\right] \quad (18)$$

where $C_I\,\Delta t$ is the salt concentration in the irrigation water.
The salt concentration of the groundwater $C_{gw}(t)$ can be estimated as:

$$C_{gw}(t) = \frac{[G(t-1)\times C_{gw}(t-1) + C(5,t)\times R_w(t)]}{G(t-1) + R_w(t)} \quad (19)$$



Where $C(5, t)$ is the soil salinity concentration of the soil layer *5* on day $t$ (g L⁻¹),
$G(t-1)$ is the difference of the groundwater depth and the depth that the largest
groundwater table fluctuations depth of groundwater table on day *(t-1)* (m) (Xue et al.,
2018), $C_{gw}(t)$ is the soluble salt concentration of groundwater at day $t$ (g L⁻¹).
2.3.2.4 Upward flux
For the upward flux period, it is assumed there is no downward water flux to
groundwater in this study. The evapotranspiration leads to the decrease of soil moisture
content in the vadose zone and lowers the groundwater table due to the upward
movement of groundwater to crop root zone and soil surface. The soil moisture content
is calculated by taking the difference of equilibrium moisture content associated with the
change of groundwater depth.
**Water**
The groundwater upward flux, $U_{gw}(h, t)$, is limited by either the maximum upward flux
of groundwater, $U_{gw,max}(h)$, or the actual evapotranspiration, formally stated as:

$$U_{gw}(h, t) = min\left[[E_a(t) + T_a(t)], U_{gw,max}(h)\right] \quad (20)$$

$$E_a(t) = \sum_{j=1}^{n} E_a(j, t) \quad (21)$$

$$T_a(t) = \sum_{j=1}^{n} T_a(j, t) \quad (22)$$

The maximum upward flux can be expressed as (Liu et al., 2019; Gardner et al., 1958)

$$U_{gw,max}(h) = \frac{a}{e^{bh} - 1} \quad \text{for } U_{gw}^h \leq ET_p \quad (23)$$

where *a* and *b* are constants that need to be calibrated.
Two cases are considered for determining the moisture contents of the layers



depending on whether the actual evapotranspiration is greater or less than the maximum
upward flux.
Case I: $U_{gw,max}(h) > E_a(t) + T_a(t)$
In this case, where the maximum upward flux is greater than the evaporative demand, the
groundwater depth is updated

$$h(t) = h(t - \Delta t) + \frac{E_a(t) + T_a(t)}{\mu(\bar{h})} \qquad (24)$$


where $\mu(\bar{h})$ is the average drainable porosity over the change in groundwater depth h.
The moisture content after the change in groundwater depth becomes

$$\theta(j, t) = \theta(j, t - \Delta t) + \theta_{fc}(j, h(t)) - \theta_{fc}(j, h(t - \Delta t)) \qquad (25)$$


Note that when the layer is at field capacity and the upward flux is equal to the
evaporative flux, the layer remains at field capacity for the updated groundwater depth at
time t.
Case II: $U_{gw,max}(h) \leq E_a(t) + T_a(t)$
In this case, the groundwater depth is updated

$$h(t) = h(t - \Delta t) + \frac{U_{gw,max}(h)}{\mu(\bar{h})} \qquad (26)$$


When the upward flux is less than the sum of the actual evaporation and transpiration,
the moisture content is updated with the difference between the two fluxes,
$U_{gw,max}(h)$ and $[E_a(t) + T_a(t)]$, according to a predetermined distribution extraction of
water out of the root zone
$\theta(j, t) = \theta(j, t - \Delta t) + \theta_{fc}(j, h(t)) - \theta_{fc}(j, h(t - \Delta t) - \dfrac{r(j)[E_a(t) + T_a(t) - U_{gw,max}(h)]}{L(j)}$  (27)
The upward flux of water can be found by summing the differences in moisture content



above the layer $j$ similar to Eq 14, but starting the summation at the groundwater.

**Salinity**

The salt from groundwater is added to the soil layers according to the root function. The

soil salinity concentration in layer $j$ at day $t$ can be expressed as

$$C(j,t) = \frac{\theta(j,t-\Delta t)\,C(j,t-\Delta t)L(j) + r(j,t)U_g(h,t)C_{gw}(t)}{\theta(j,t-\Delta t)L(j)+(\theta_{fc}(j,h(t)) - \theta_{fc}(j,h(t-\Delta t)))L(j) - r(j,t)(E_a(t) + T_a(t) - U_{gw,max}(h))} \quad (28)$$

Since water is extracted from the reservoir that has the same concentration as in the

reservoir, the concentration will not change, hence the equation used to estimate the

groundwater salt concentration can be expressed as

$$C_{gw}(t) = C_{gw}(t - \Delta t) \qquad (29)$$

## 3. Data collection

3.1 Study area

Field experiments were conducted in 2017 and 2018 in Shahaoqu experimental station

in Jiefangzha sub-district, Heato irrigation district in Inner Mongolia, China (Fig. 3).

Irrigation water originates from the Yellow River. The area has an arid continental climate.

Mean annual precipitation is 155 mm a$^{-1}$ of which 70% falls from June to September. Pan

evaporation is 2000 mm a$^{-1}$ (Xu et al., 2010). The mean annual temperature is 7°C. The

soils begin to freeze in the middle of November and to thaw in end of April or beginning

of May. Maize, wheat and sunflower are the main crops in Jiefangzha sub-district and are

grown with flood irrigation. The groundwater depth is between 0.5-3 m. Regional

exchange of groundwater is minimal due to low gradient of 0.01-0.025 (Xu et al., 2010).

Thus, the groundwater mainly moves in a vertical direction in the regional scale. Soil





salinity in the aquifer in over 86% of the Hetao district is less than 2 g L$^{-1}$.

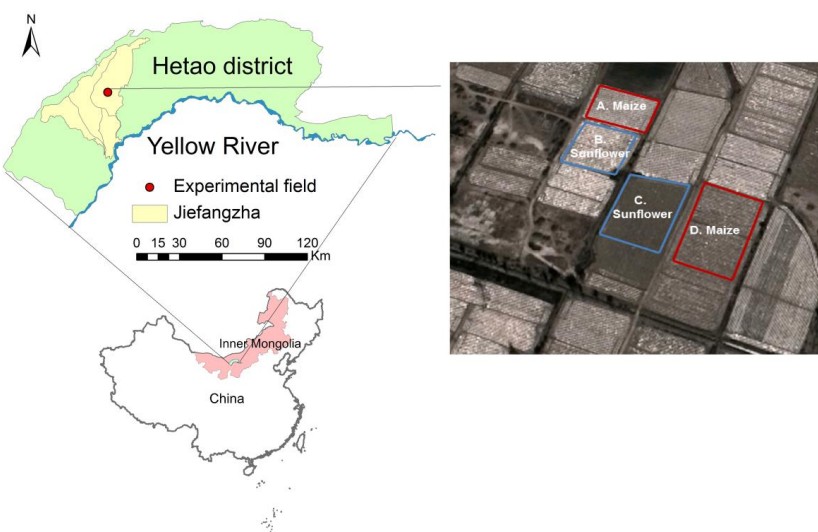


Fig. 3 Location of the Shahaoqu experimental field (Note: The figure about the layout of

the experimental fields is download from © Google earth)


3.2 Field observations and data

The layout of the experimental fields is shown in Figure 3. The areas of fields A, B, C

and D are 920, 2213, 1167, 1906 m$^2$, respectively. Field A and D were planted with
maize on May 10 and harvested on September 30, 2017. In 2018, fields A and D were
planted with gourds and were therefore not monitored in 2018. Fields B and C were
seeded with sunflower in both 2017 and 2018. The sunflower was planted on June 1,
2017 and June 5, 2018. Harvest was on September 15 in both years. The fields were
flood irrigated ranging from two to five times during the growing season (Table 1). A well
was installed in each experimental field to monitor the groundwater depth.




Table 1 Irrigation scheduling for the Shahaoqu experimental fields in 2017 and 2018

| Field | Year | Irrigation events | Date | Irrigation depth (mm) |
|---|---|---|---|---|
| A (maize) | 2017 | 1 | 5/30 | 100 |
| | | 2 | 6/25 | 162 |
| | | 3 | 7/14 | 275 |
| | | 4 | 8/6 | 199 |
| B (sunflower) | 2017 | 1 | 6/26 | 140 |
| | | 2 | 7/23 | 121 |
| | 2018 | 1 | 6/20 | 134 |
| | | 2 | 6/24 | 60 |
| | | 3 | 7/15 | 114 |
| | | 4 | 7/22 | 40 |
| | | 5 | 8/31 | 130 |
| C (sunflower) | 2017 | 1 | 6/19 | 80 |
| | | 2 | 6/30 | 80 |
| | 2018 | 1 | 6/20 | 140 |
| | | 2 | 7/14 | 100 |
| D (maize) | 2017 | 1 | 6/13 | 150 |
| | | 2 | 6/26 | 94 |
| | | 3 | 7/6 | 50 |
| | | 4 | 7/14 | 174 |
| | | 5 | 8/6 | 120 |


Daily meteorological data, including air temperature, precipitation, relative humidity,
wind speed, and sunshine duration, originated from the weather station at the Shahaoqu
experimental station. The soil moisture content for the four experimental fields in 2017
and for field C in 2018 during the crop growing season was measured every 7-10 days
at the depths of 0-20, 20-40, 40-60, 60-80, 80-100 cm by taking soil samples and
oven drying. For field B in 2018, the soil moisture content was monitored daily in the top
100cm at 20 cm intervals using Hydra Probe Soil Sensors (Stevens Water Monitoring



System Inc., Portland, OR, USA). In 2017, the groundwater depths were manually
measured in all four experimental fields about every 7-10 days. In 2018, the
groundwater depth in fields B and C was recorded at 30 min intervals using an HOBO
Water Level Logger-U20 (Onset, Cape Cod, MA, USA). The sensors of the soil moisture
content and groundwater depth were connected to data loggers and downloaded via
wireless transmission. The crop leaf area and crop height were manually measured every
7-12 days.

Undisturbed soil samples were collected in 5 cm high rings with a diameter of 5.5

cm from the five soil layers where the soil moisture were taken and used for textual
analysis, saturated soil moisture content, field capacity and soil bulk density. The soil
texture was analyzed with a laser particle size analyzer (Mastersizer 2000, Malvern
Instruments Ltd., United Kingdom). The American soil texture classification method was
used in this study. Finally, the soil samples were collected 7-10 days apart to monitor the
change of electrical conductivity (EC). The soil samples were mixed with distilled water in
a proportion of 1:5 to measure the electrical conductivity of the soil water by a portable
conductivity meter. It is assumed that 1 ms cm$^{-1}$ corresponds to 640 mg L$^{-1}$ of total
dissolved salts (Wallender and Tanji, 2011; Xue et al., 2018).
3.3 Model calibration and validation
The observed soil moisture contents, groundwater depths, crop heights, LAIs and salinity
concentrations for field A with maize and sunflower fields B and C in 2017 were used for
calibration and the sunflower data of fields B and C in 2018 and the maize data in field
D in 2017 were used for validation. The initial $\vartheta_{0.33}$ was based on the measured data



(Table 2). The initial values of $\vartheta_s$ and $\vartheta_{15}$ were derived from the soil texture with the
method of Ren et al. (2016) (Table2). The default values of EPIC for sunflower and maize
were used as initial values for simulating crop growth ($K_{cmax}$ and $LAI_{mx}$ in Eq. S3, $K_b$ in Eq.
S4, $H_{mx}$ in Eq. S7, $PHU$ in Eq. S9, $T_b$ in Eq. S10, $ad$ in Eq. S12, $T_O$ and $T_b$ in Eq. 16, $RD_{mx}$ in
Eq. S18). The initial value maximum crop coefficient ($K_{cmax}$) in Eq. S3 in Supplementary S1
for evapotranspiration calculation was taken from *Sau et al.*, (2004). The initial values of
two groundwater parameters (*a* and *b* in Eq. 23) were based on Liu et al., (2019). The
Brooks and Corey soil moisture characteristic parameters ($\varphi_b$, $\lambda$ in Eq. 8) were obtained
by fitting the outer envelope of the measure moisture content and water table data.
Statistical indicators were used to evaluate goodness of fit of the hydrological model
for both calibration and validation (Ritter and Muñoz-Carpena, 2013). The statistical
indicators included the root mean square error (RMSE) (Abrahart and See, 2000),
$$\text{RMSE} = \sqrt{\frac{1}{N}\sum_{i=1}^{N}(P_i - O_i)^2} \qquad (30)$$

the mean relative error (MRE) (Dawson et al., 2006; Nash and Suscliff, 1970),
$$\text{MRE} = \frac{1}{N}\sum_{i=1}^{N}\frac{(P_i - O_i)}{O_i} \times 100\% \qquad (31)$$

the Nash-Sutcliffe efficiency coefficient (NSE) (Nash and Suscliff, 1970),
$$\text{NSE} = 1 - \frac{\sum_{i=1}^{N}(P_i - O_i)^2}{\sum_{i=1}^{N}(O_i - \bar{O})^2} \qquad (32)$$

and the determination coefficient (R$^2$) and regression coefficient (b) (Xu et al., 2015)
$$\text{R}^2 = \left[\frac{\sum_{i=1}^{N}(O_i - \bar{O})(P_i - \bar{P})}{[\sum_{i=1}^{N}(O_i - \bar{O})]^{0.5}[\sum_{i=1}^{N}(P_i - \bar{P})]^{0.5}}\right]^2 \qquad (33)$$



$$b = \frac{\sum_{i=1}^{N} O_i \times P_i}{\sum_{i=1}^{N} O_{i=1}^2} \qquad (34)$$

where $N$ is the total number of observations; $P_i$ and $O_i$ are the $i^{th}$ model predicted and
observed values ($i$=1,2,3…N), respectively; $\bar{O}$ and $\bar{P}$ are the mean observed values and
predicted values, respectively. The value of RMSE and MRE close to 0 indicates good
model performance. The value of NSE ranges from -∞ to 1. NSE=1 means a perfect fit
while the negative NSE value indicates the mean observed value is a better predictor
than the simulated value (Moriasi et al., 2007). For b and $R^2$, the value closest to 1
indicates good model predictions.
3.4 Parameters sensitivity analysis
A sensitivity analysis was performed to determine how the input parameters
affected output of the models (Cloke et al., 2008; Cuo et al., 2011). Each parameter was
varied over a range of -30% to 30% to derive the corresponding impact on the model
output. The change in output values was plotted against the change in input values.

**4 Results**
The 2017 and 2018 experimental data of the Shahaoqu farmers' fields in the Hetao
irrigation district (Fig.3) are presented first, followed by the calibration and validation of
the CROP and VADOSE modules of EPICS model.
4.1 Results of the field experiment
4.1.1 Water input
The precipitation was 63 mm in 2017 (May 10 to September 30) and 108 mm in
2018 (June 1 to September 15). The precipitation from the greatest rainstorm was 26



mm on September 1, 2018 (Fig. 4). Irrigation provided most of the water for the crops.
Field A (maize) was irrigated four times with a total of 736 mm and field D (maize) was
irrigated five times for a total of 588 mm in 2017. Sunflower fields B and C were both
irrigated twice with a total water amount of 261mm and 160mm, respectively, in 2017.
In 2018, fields B and C were irrigated five and two times, respectively, with a total water
amount of 478mm and 240mm, respectively. The total reference evapotranspiration
from May 10 to September 30, 2017 was 595 mm and 368 mm from June 1 to
September 15, 2018. On a daily basis, the reference evapotranspiration ranged from 1
mm d$^{-1}$ to a maximum of 6.4 mm d$^{-1}$ during crop growth period (Fig. 4).

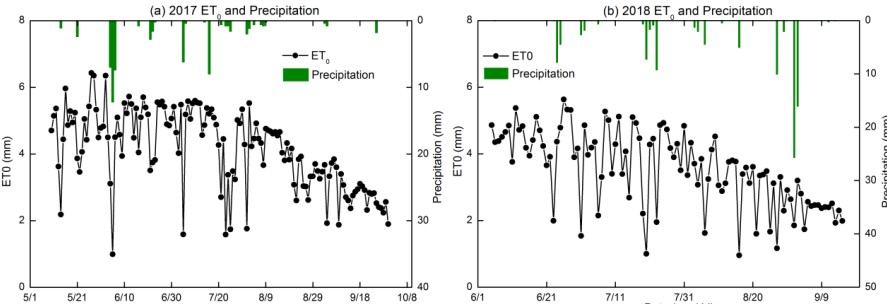


Fig 4. Reference evapotranspiration (ET$_0$) and precipitation during crop growth period in
2017 and 2018.

4.1.2 Soil physical properties
Based on the soil textural analysis in Table 2, the soils were classified as silt, silt loam and
loamy sand. Bulk densities varied from 1.24 to 1.47 Mg m$^{-3}$ with the greatest bulk
densities in the 0-20 cm soil layer. There was generally more sand in the top 40 cm than
below. The subsoil was heavier and had the greatest percentage of silt (Table 2). The
moisture content at -33 kPa (0.33 bar) varied from 0.25 to 0.35 cm$^3$cm$^{-3}$ and at 1.5Mpa





(wilting point at15 bar) ranged from 0.08 to 0.15 cm³cm⁻³ (Table 2).
Table 2 Soil texture and bulk density of the experimental fields in Shahaoqu

| Field | Soil depth (cm) | Sand(%) | Silt(%) | Clay(%) | Soil type | $\rho$(Mg m⁻³) | $\theta_{0.33}$(m³m⁻³) | $\theta_{15}$(m³m⁻³) |
|---|---|---|---|---|---|---|---|---|
| A | 0-20cm | 26 | 62 | 13 | Silt loam | 1.44 | 0.31 | 0.1 |
| | 20-40cm | 76 | 22 | 2 | Loamy sand | 1.24 | 0.32 | 0.07 |
| | 40-60cm | 10 | 79 | 10 | Silt loam | 1.33 | 0.33 | 0.12 |
| | 60-100cm | 6 | 79 | 15 | Silt loam | 1.35 | 0.34 | 0.14 |
| | | | | | | | 0.35 | 0.14 |
| B | 0-20cm | 22 | 64 | 13 | Silt loam | 1.44 | 0.29 | 0.15 |
| | 20-40cm | 16 | 73 | 11 | Silt loam | 1.24 | 0.26 | 0.13 |
| | 40-60cm | 18 | 73 | 9 | Silt loam | 1.33 | 0.32 | 0.11 |
| | 60-80cm | 8 | 77 | 16 | Silt | 1.35 | 0.34 | 0.14 |
| | 80-100cm | 13 | 79 | 8 | Silt loam | | 0.35 | 0.12 |
| C | 0-20cm | 29 | 63 | 8 | Silt loam | 1.47 | 0.26 | 0.08 |
| | 20-40cm | 37 | 56 | 6 | Silt loam | 1.33 | 0.25 | 0.08 |
| | 40-60cm | 35 | 59 | 7 | Silt loam | 1.32 | 0.26 | 0.08 |
| | 60-80cm | 14 | 74 | 12 | Silt loam | 1.38 | 0.31 | 0.12 |
| | 80-100cm | 10 | 82 | 8 | Silt | 1.38 | 0.34 | 0.11 |
| D | 0-20cm | 27 | 62 | 11 | Silt loam | 1.47 | 0.3 | 0.15 |
| | 20-40cm | 5 | 80 | 15 | Silt loam | 1.33 | 0.27 | 0.14 |
| | 40-60cm | 7 | 75 | 18 | Silt loam | 1.32 | 0.33 | 0.15 |
| | 60-100cm | 10 | 81 | 9 | Silt | 1.38 | 0.34 | 0.12 |
| | | | | | | | 0.31 | 0.14 |


4.1.3 Soil moisture content

Moisture content, rainfall and irrigation amounts are depicted for the five layers and

the four fields in 2017 and two fields in 2018 in Fig. 5. Blue closed spheres indicate that
the moisture content was determined on cored soil samples (Figs. 5a, b, c, e, f) and
close-spaced spheres when the hydra probe was used (Fig. 5d). The moisture contents
were near saturation when irrigation water was added and subsequently decreased due
to crop transpiration and soil evaporation (Fig. 5). In all cases, the moisture contents



during the main growing period remained above the moisture content at -33 kPa that
ranged from 0.25 cm³cm⁻³ to 0.34 cm³cm⁻³ for the 60-80 cm depth (Table 2, Fig.5). Only
after the last irrigation and during harvest of the crop did the moisture content in the top
0-40 cm for maize and 0-60 cm for sunflower decrease below the moisture content at
-33kPa. During the growing season, the variation of moisture content was greater in the
top 60 cm with the majority of the roots than in the lower depths where, after the first
irrigation, it remained nearly constant close to saturation.

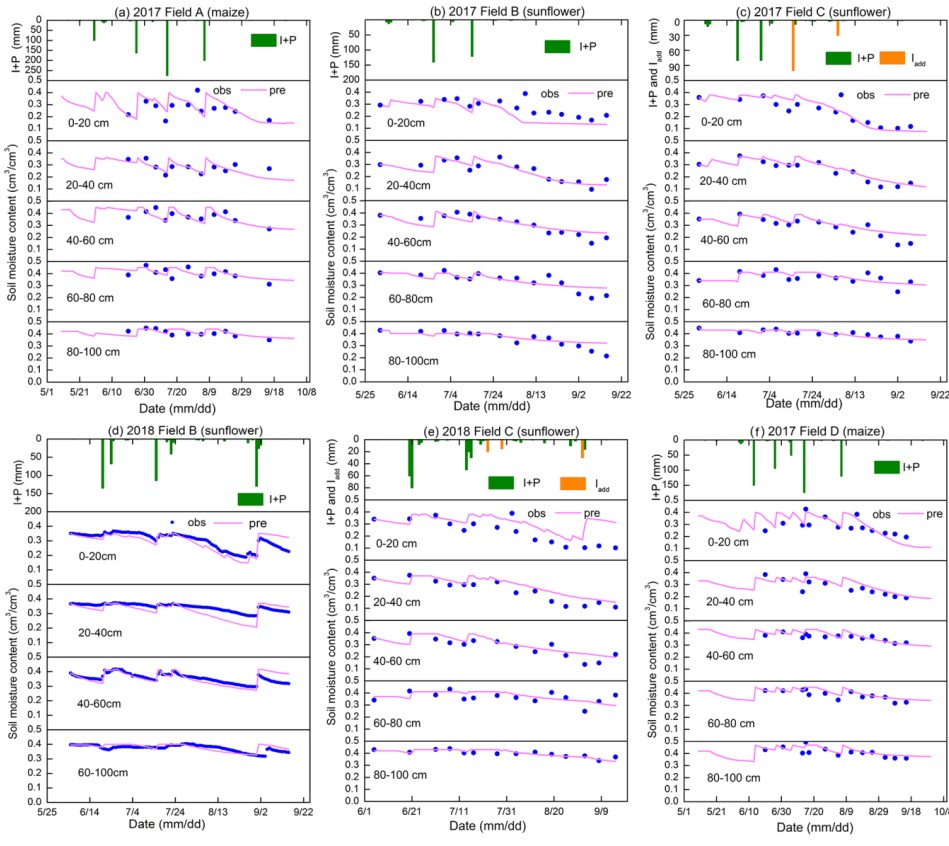


Fig. 5 Observed (blue dots) and simulated soil moisture content of the Shahaoqu
experimental fields during model calibration (a,b,c) and validation (d,e,f)



4.1.4 Salinity

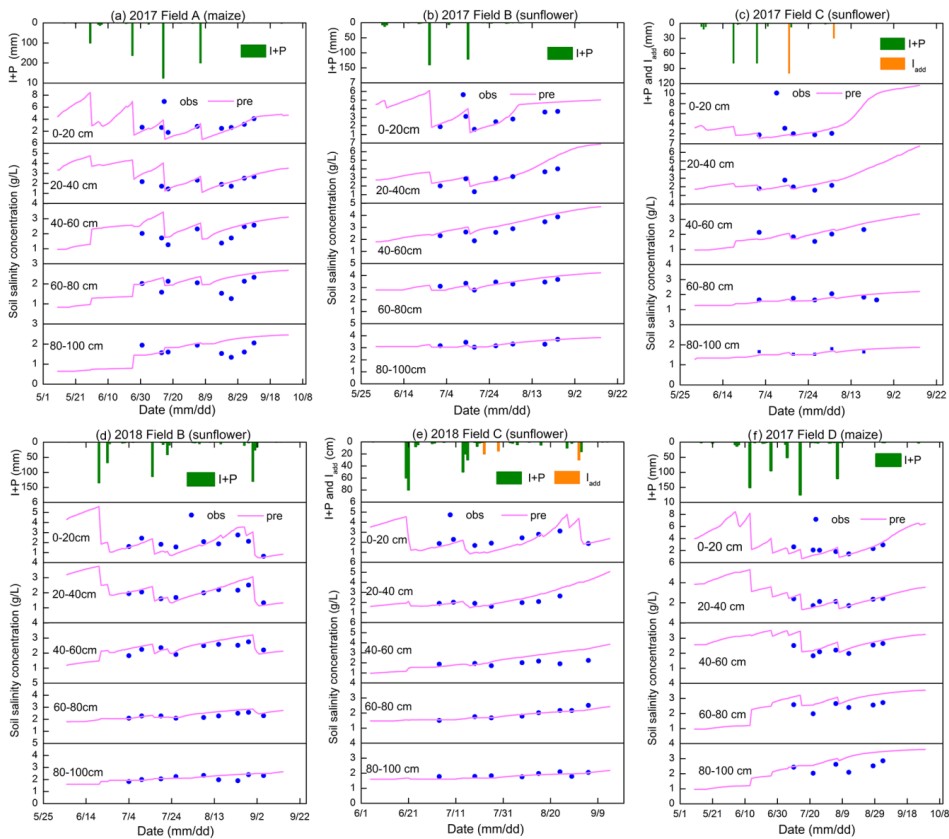

Fig. 6 Observed (blue dots) and simulated soil salinity concentration of the experimental
fields in Shahaoqu during model calibration (a,b,c) and validation (d,e,f).

Overall the salt concentration is greatest at the surface and increases at all depths

during the growing season. Sunflower is more salt tolerant than maize and the overall
salt concentration was greater in the sunflower fields (Fig. 6) at comparable times of the
crop development for field B but not for field C. Comparing the salt concentration and
soil moisture patterns (Fig.5), we note that they behave similarly but opposite to each
other (Fig. 6). The soil salinity concentration was decreasing during an irrigation event



due to dilution and then gradually increasing partly due to evaporation of the water.
Some of the soil salt was transported to the layers below during irrigation and some salt
was moving upward with the evaporation from the surface. As expected, after the harvest,
the autumn irrigation decreased the salt concentration from fall 2017 to spring 2018.

4.1.5 Groundwater observations
The variation in groundwater depth during the growing season was very similar for
both years and in all fields. The groundwater depth for all fields was between 50 and
100 cm from the surface after an irrigation event and then decreased to around 150 cm
before the next irrigation or rainfall (Fig.7). Only after the last irrigation in August 2017
did the water table decrease to below 250 cm and to around 200 cm in 2018. Field D
followed the same pattern but the groundwater was more down from the surface. In
several instances, the groundwater table increased without an irrigation or rainfall event
in sunflower field C (Fig. 7c and 7e). This was likely related to an irrigation event either
from a spillover or an accidental irrigation that was not properly documented. We
estimated the amount of irrigation water based on the change in moisture content in the
soil profile (orange bars in Fig. 7c and 7e). Finally, there was a notable rise in the water
table of an mean 375mm "autumn irrigation" after harvest between the end of 2017
(Figs. 7 a, b, c) and the beginning of 2018 (Figs. 7 d, e, f) ,which is a common practice in
the Jiefangzha irrigation district to leach the salt that has accumulated in the profile
during the growing periods.
Note that in Fig. 7, after an irrigation event, the groundwater depth was between





50-80 cm while the whole profile was saturated (Fig. 5). This is directly related to the
bubbling pressure of the water. After the irrigation event stopped, the water table was
likely at the surface but then immediately decreased because a small amount of
evaporated water will bring the water table down to a depth of approximately equal to
the bubbling pressure, $\varphi_b$, in Eq. 5. The bubbling pressures are listed in Table 3.
4.1.6 LAI and plant height

Plant height and LAI followed the typical growth curve that started slowly to rise in

the beginning, accelerated during the vegetative stage and then became constant during
the seed setting and ripening stages (Fig. 8). In the maturing stage, the leaf area index
decreased.

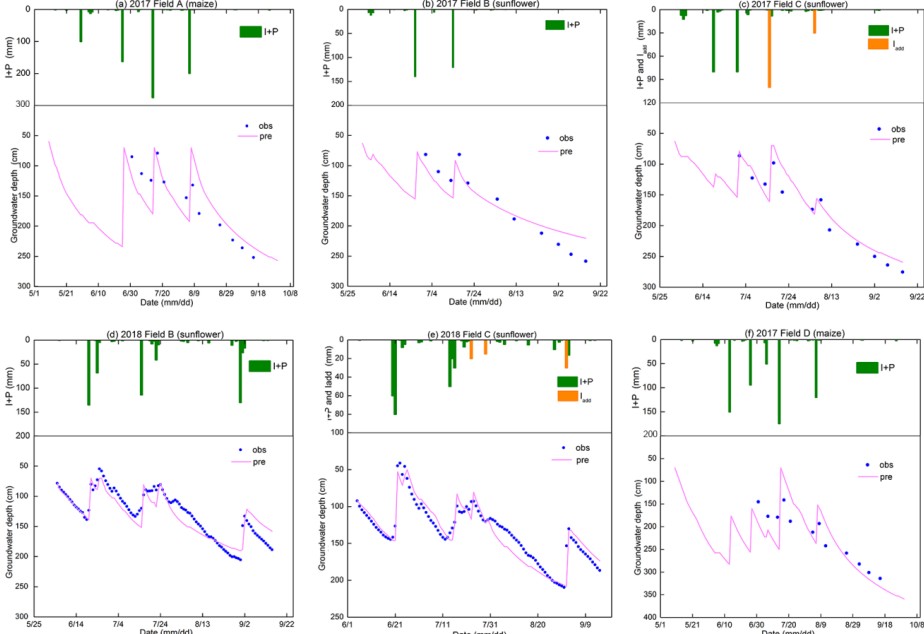


Fig. 7. Observed (blue dots) and simulated groundwater depth of the experimental fields
in Shahaoqu during model calibration (a, b, c) and validation (d, e, f)





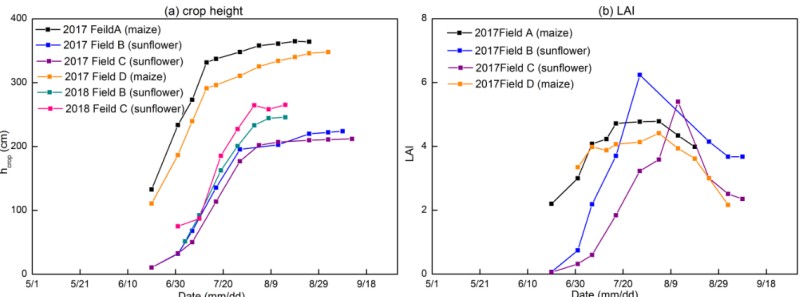

Fig. 8 Observed crop height (a) and leaf area index (b) of the experimental field in
Shahaoqu in 2017 and 2018.

4.2 Soil Characteristic curve and drainable porosity

To simulate the soil moisture content and to derive drainable porosity as a function of

water table depth, the soil moisture characteristic curves were derived by plotting the

observed soil moisture content in 2017 and 2018 versus the height above the water

table to the soil surface for the five soil layers in Fig. 9. The Brooks-Corey equation

(Brooks and Corey, 1964) was fitted through outer envelope of the points. The

parameters of the Brooks-Corey equation were adjusted through a trial and error to

obtain the best fit (Table 3a). In Fig. 9, points on the left side of the soil moisture

characteristic curve (moisture content smaller than the field capacity) were due to water

removal at times when evaporative demand was greater than the upward water flux. The

few points at the right of the soil moisture characteristic curve indicate the soil moisture

was greater than field capacity and matric potential and groundwater were not yet at

equilibrium after an irrigation event.

The fitted parameter values are consistent. Field A had a greater bubbling pressure

and moisture content at -33 kPa than the other fields indicating that this field had more



clay. This was confirmed by the data in Table 2. For fields B, C and D, the bubbling
pressure was greater at the 60-80 cm depth or the 80 -100 cm depth, which was also in
accordance with the data in Table 2.
Table 3a Calibrated soil hydraulic parameters in the Brooks and Corey soil moisture
characteristic curve.

| Field | Parameter | 0-20cm | 20-40cm | 40-60cm | 60-80cm | 80-100cm |
|---|---|---|---|---|---|---|
| A | $\theta_s$ | 0.4 | 0.36 | 0.43 | 0.45 | 0.47 |
| | $\varphi_b$ | 80 | 100 | 90 | 70 | 50 |
| | $\lambda$ | 0.18 | 0.21 | 0.22 | 0.18 | 0.15 |
| B | $\theta_s$ | 0.35 | 0.37 | 0.41 | 0.4 | 0.4 |
| | $\varphi_b$ | 50 | 55 | 33 | 60 | 55 |
| | $\lambda$ | 0.14 | 0.11 | 0.16 | 0.2 | 0.2 |
| C | $\theta_s$ | 0.38 | 0.37 | 0.39 | 0.71 | 0.43 |
| | $\varphi_b$ | 55 | 50 | 40 | 60 | 40 |
| | $\lambda$ | 0.26 | 0.24 | 0.2 | 0.18 | 0.13 |
| D | $\theta_s$ | 0.4 | 0.36 | 0.45 | 0.45 | 0.44 |
| | $\varphi_b$ | 50 | 40 | 55 | 50 | 50 |
| | $\lambda$ | 0.21 | 0.2 | 0.3 | 0.17 | 0.15 |

Note: $\theta_s$ is the soil moisture at saturation ($cm^3cm^{-3}$), $\varphi_b$ is bubbling pressure (cm), $\lambda$ is the
pore size distribution index.
Table 3b Calibrated groundwater parameters

| Field\parameters | A | B | C | D |
|---|---|---|---|---|
| a | 70 | 75 | 110 | 70 |
| b | 0.02 | 0.025 | 0.022 | 0.015 |


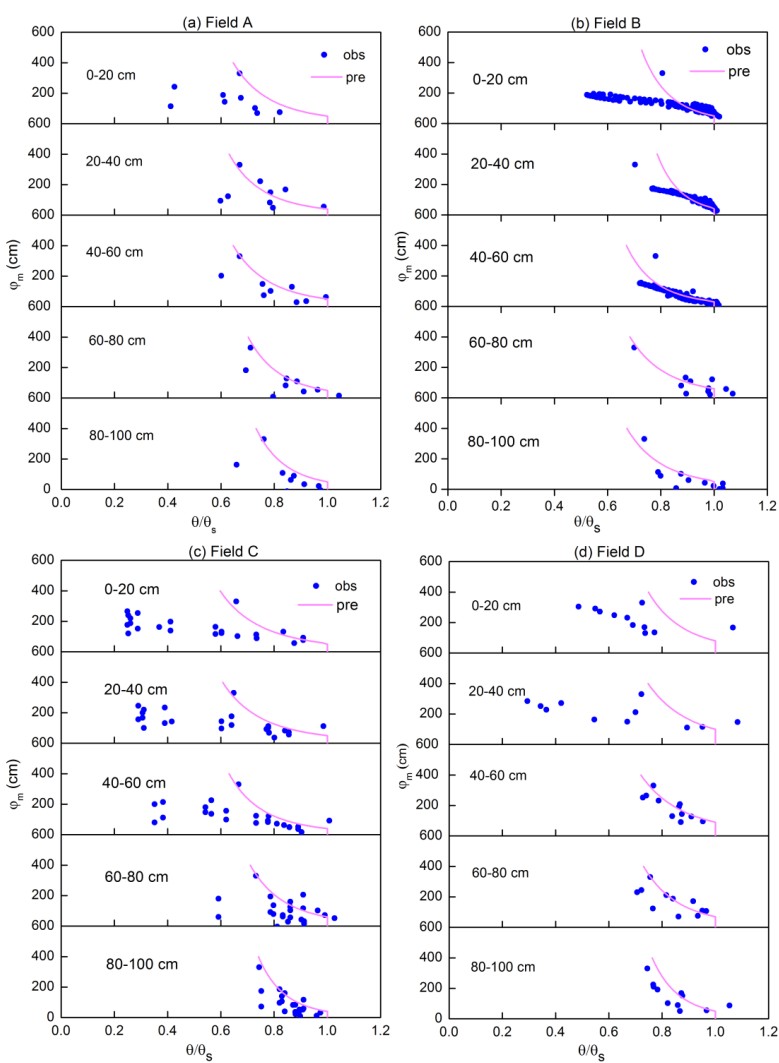

Figure. 9 Soil moisture characteristic curves of five soil layers in the experimental fields. The red line is the fit with the Brooks-Corey equation.

## 4.3 Parameters sensitivity analysis

The results of sensitivity analysis of the 15 input parameters on 5 output parameters are shown in Fig. 10. The evaluated output parameters are soil moisture content, groundwater depth, soil salinity concentration, field evapotranspiration, and crop leaf

area index (LAI). Steeper lines indicate a greater sensitivity of the parameter.
The results of the sensitivity analysis show that moisture content predictions (Fig
10a) are the most sensitive to the input value of the saturated moisture content ($\vartheta_s$).
None of the other parameters are very sensitive. The input parameter with the most
sensitivity for *groundwater depth (*Fig. 10b), is the saturated moisture content as well.
Other less sensitive parameters are the exponent *b* and constant *a* in Eq. 23 in predicting
the upward flux and the bubbling pressure, $\varphi_b$, of the soil moisture characteristic curve
(Eq. 8a). Likewise, in case of the *salinity* predictions (Fig. 10c), the saturated moisture
content gives the greatest relative change in salt content. Less sensitive, but still
important, are the field capacity, $\theta_{0.33}$, the bubbling pressure, $\varphi_b$, and the exponent $\lambda$ of
the soil characteristic curve (Eq. 8a) and *b* in Eq. 23. The sensitive parameters for the *leaf*
*area index (LAI)* (Fig 10d) are the maximum potential leaf area index, $LAI_{mx}$ and fraction
of growing season when leaf area declines (*DLAI*) followed by total potential heat units
required for crop maturation (*PHU*). Finally, for the evapotranspiration (Fig 10e), the
saturated soil moisture content is the most sensitive parameter, and other less sensitive
parameters are the exponent *b* and field capacity.
Thus, the model output is most sensitive to the input parameters that define the soil
hydraulic properties, groundwater flux and crop growth. As expected, since the soil
remains near field capacity, the parameters that relate to the reduction of evaporation
when the soil dries out are insensitive. When used in the simulation practices, the model
needs to be calibrated and verified to avoid high error from parameters uncertainty.



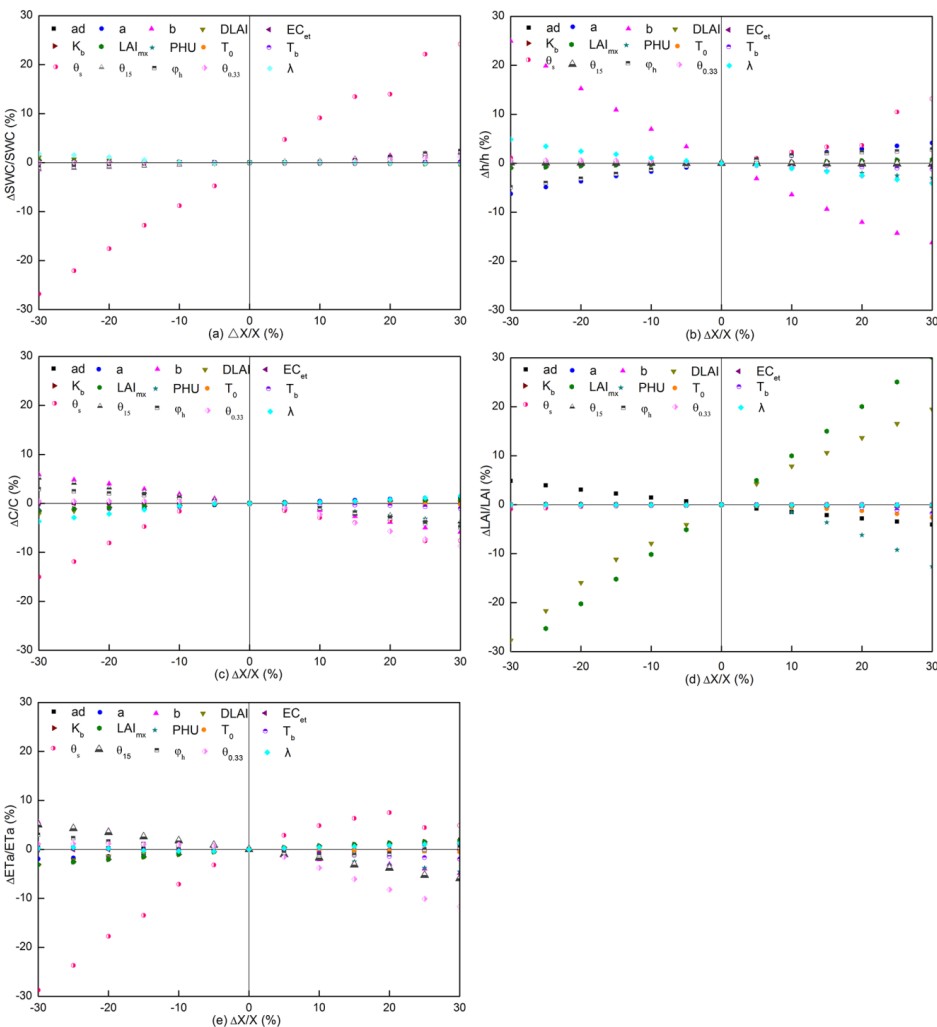


Figure. 10 Parameters sensitivity analysis for (a) soil moisture content, (b) groundwater
depth, (c) salt salinity concentration, (d) LAI, (e) ET

**4.4 Model calibration and validation**
The model parameters were calibrated and validated using the observed moisture
contents, groundwater depth, plant height, leaf area index and the calculated
evapotranspiration. For calibration, the data collected in 2017 were used for sunflower



fields B and C and maize field A. Since farmers did not grow maize in 2018, the 2017
data of maize field D, together with sunflower fields B and C in 2018 were used for
validation. The optimal parameter set was determined using graphical similarity between
observed and predicted results together with near optimum performance of the
statistical indicators while keeping all values within physical acceptable ranges.

As a way of reducing the number of parameters that needed to be calibrated, we

initially selected one to three most sensitive parameters for each of the observed time
series, starting with evapotranspiration (including $LAI$ and crop height) followed by
moisture content, groundwater depth, and salt content in the soil. This cycle was
repeated several times until changes became small. The last stage of the calibration
consisted of fine-tuning the remaining least sensitive parameters.

To calibrate the parameters in the CROP module, we calculated evapotranspiration

during the crop growth period with the observed soil moisture content and groundwater
depth by the soil water balance method. In addition, we used the observed $LAI$
measurements in 2017 and plant height in both 2017 and 2018. $LAI$ was not measured
in 2018. The $DLAI$, $LAI_{mx}$ and $H_{mx}$ in the crop module were adjusted to fit the observed
$LAI$ and crop height values. In addition, we fitted the $\theta_{0.33}$ moisture content to obtain a
good fit of the evapotranspiration. The saturated moisture content values were not
adjusted since they were already determined for fitting the soil characteristic curve. The
exponent $b$ and constant $a$ in Eq. 23 were adjusted to fit the observed soil moisture
content and groundwater depth.
*4.4.1 Evapotranspiration, crop height and leaf area index*



The predicted evapotranspiration and that calculated from the mass balance show a
good agreement with Nash Sutcliff values ranging from 0.96-0.89 during calibration and
validation (Fig. 11 and Table 4). The calibrated predictions of plant height fitted the
observed values well during calibration and validation with Nash Sutcliff values ranging
from 0.77-0.96 for the individual fields (Table 4) and over 90% when the data was
pooled for the fields during calibration and validation (Fig.12). LAI was not measured in
2018. During calibration, Nash Sutcliff predicted LAI values were good for sunflower but
not as good for maize but the coefficient of determination and slope in the regression
were acceptable (Table 4, Fig. 13). In addition, the overall trend was predicted reasonably
well (Fig. 13b).

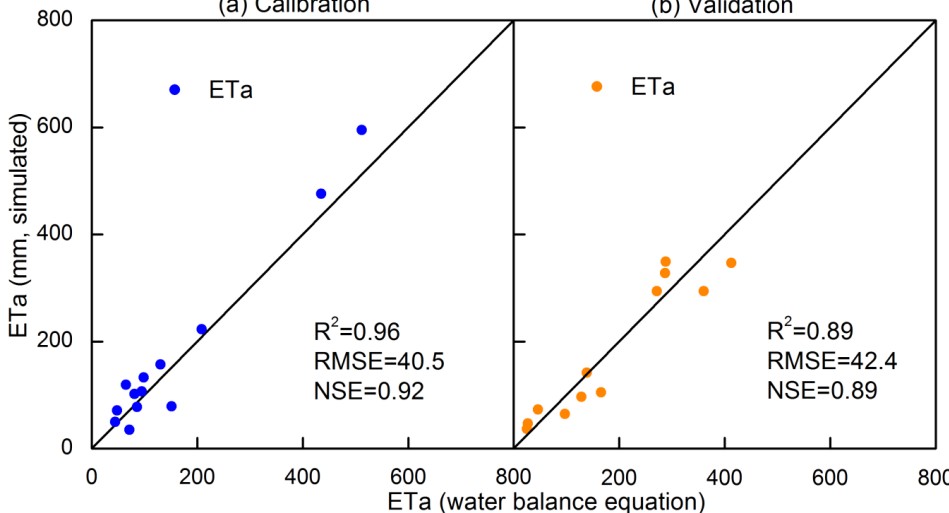


Fig. 11 Comparison of predicted and observed actual evapotranspiration: a) Calibration
and b) Validation





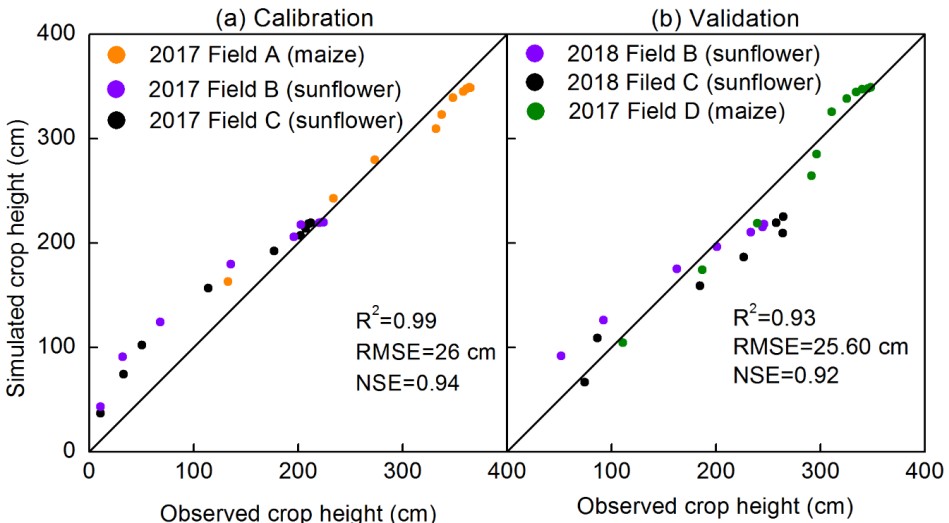

Fig.12 Comparison of predicted and observed crop height: a) Calibration and b) Validation

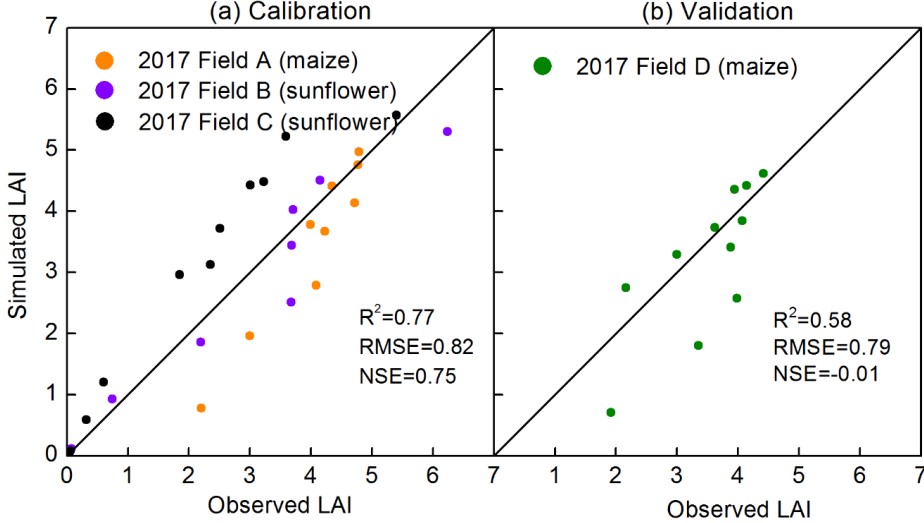

Fig. 13 Comparison of predicted and observed LAI: a) Calibration and b) validation





Table 4 Model error statistics for calibration and validation of model in 2017 and 2018
(Mean relative error, MRE; root mean square error, RMSE; Regression slope; Coefficient of
determination, $R^2$; Regression coefficient, slope).

| Process | Field | Variable | MRE (%) | RMSE ($cm^3cm^{-3}cm$ or gL-1or mm) | NSE | $R^2$ | Regression coefficientslope |
|---|---|---|---|---|---|---|---|
| Calibration | 2017 Field A (maize) | SWC (0-1m) | 2.9 | 0.04 | 0.8 | 0.56 | 1.01 |
| | | GWD | 4.5 | 33.8 | 0.64 | 0.64 | 0.97 |
| | | LAI | -17.4 | 0.78 | 0.11 | 0.92 | 0.89 |
| | | hcrop | 0.04 | 16.2 | 0.95 | 0.99 | 0.97 |
| | | C | 13.9 | 0.5 | 0.27 | 0.49 | 1.07 |
| | 2017 Field B (sunflower) | SWC (0-1m) | -1.2 | 0.04 | 0.71 | 0.74 | 0.97 |
| | | GWD | 6.0 | 22.9 | 0.86 | 0.98 | 0.96 |
| | | LAI | 4.7 | 0.58 | 0.9 | 0.92 | 0.91 |
| | | hcrop | 6.8 | 33.5 | 0.83 | 0.96 | 1.1 |
| | | C | 11.0 | 0.55 | 0.27 | 0.7 | 1.1 |
| | 2017 Field C (sunflower) | SWC (0-1m) | 8.5 | 0.04 | 0.88 | 0.9 | 1.05 |
| | | GWD | -7.3 | 19.1 | 0.91 | 0.94 | 0.94 |
| | | LAI | 48.6 | 1.0 | 0.59 | 0.93 | 1.29 |
| | | hcrop | 5.42 | 27.4 | 0.88 | 0.98 | 1.07 |
| | | C | -1.6 | 0.52 | -0.64 | 0.08 | 0.94 |
| | | ETa | 12.2 | 40.5 | 0.92 | 0.96 | 1.11 |
| Validation | 2018 Field B (sunflower) | SWC (0-1m) | -2.3 | 0.03 | 0.43 | 0.68 | 0.98 |
| | | GWD | 4.86 | 16.1 | 0.83 | 0.84 | 1.01 |
| | | hcrop | 12.5 | 26.9 | 0.86 | 0.99 | 0.95 |
| | | C | 4.0 | 0.35 | 0.18 | 0.72 | 1.06 |
| | 2018 Field C (sunflower) | SWC (0-1m) | 17.3 | 0.06 | 0.64 | 0.72 | 1.04 |
| | | GWD | 2.1 | 13.8 | 0.86 | 0.87 | 1.01 |
| | | hcrop | -10.3 | 36.4 | 0.77 | 0.97 | 0.84 |
| | | C | 0.51 | 0.33 | 0.11 | 0.73 | 1.02 |
| | 2017 Field D (maize) | SWC (0-1m) | 6.1 | 0.04 | 0.68 | 0.77 | 1.05 |
| | | GWD | 0.64 | 39.1 | 0.52 | 0.71 | 1.01 |
| | | LAI | -10.7 | 0.79 | -0.02 | 0.58 | 0.93 |
| | | hcrop | -1.7 | 13.6 | 0.96 | 0.98 | 1 |
| | | C | 9.8 | 0.51 | -1.11 | 0.54 | 1.11 |
| | | ETa | 8.0 | 42.4 | 0.89 | 0.89 | 0.95 |

Note: SWC is the soil moisture content, GWD is the groundwater depth, LAI is the leaf
area index, hcrop is the height of the crop, C is the soil salinity concentration, ETa is the
actual evapotranspiration.



*4.4.2 Soil moisture and groundwater depth*
Next, the moisture contents and groundwater table were fitted with the parameters in the
Vadose model without changing the parameters in the CROP module. Saturated moisture
content was the most sensitive parameter for calibrating the moisture content (Fig.10a).
Since this value was already determined a priori from the soil characteristic curve (Table
3a), we could not use other parameters to obtain a better fit since none were sensitive
(Fig.10a). Therefore, we calibrated the groundwater parameters (i.e., *a* and *b* parameters
(Eq. 23)) together with the moisture content to obtain the best fit for both. The fitted *a*
and *b* values are listed in Table 3b. The fitted parameters between the four experimental
fields were similar but not the same. This can be expected in river plains where soils can
vary over short distances.

Overall, the moisture contents were predicted well during calibration and validation

(Figs. 5, 14 and Table 4) with the exception of field B during validation (Table 4) with a
NSE of 0.43. The moisture contents were predicted most accurately in the layers from
40-100cm where the soil moistures were at field capacity during most of the growing
season (Fig. 14). In the top 40 cm, the predicted soil moisture content deviated from
observed moisture contents, especially at the dryer end (Fig. 5 and 14). Unlike at deeper
depths, evapotranspiration determined the moisture contents at shallow depths.
Prediction of evapotranspiration introduced additional uncertainties such as the
distribution of the root system. This uncertainty is also likely the reason why the 2018
moisture contents during the validation are acceptable but not predicted as well as in

2017.

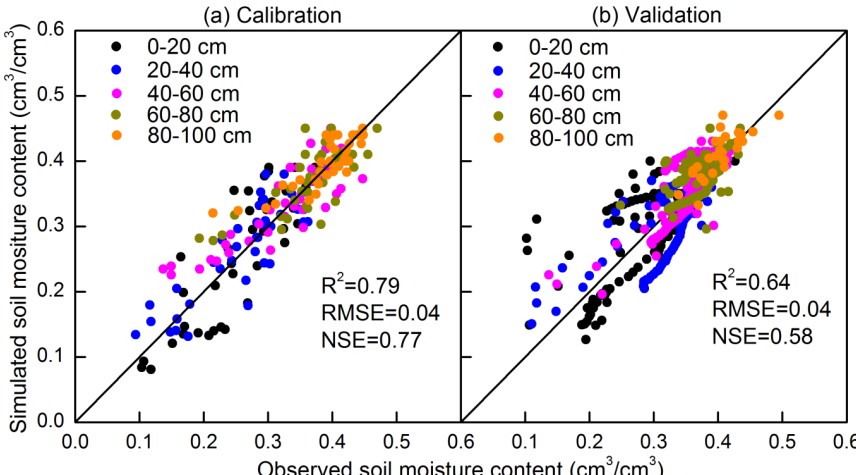


Fig. 14 Comparison of predicted and observed soil moisture content: a) calibration and b)

validation

The predicted and observed groundwater depths are in good agreement during both
calibration and validation (Figs 7, 15). The MRE values were within $\pm 10\%$ and the NSE
values ranged from 0.52 for field D during validation to 0.91 in field C during calibration
where some of the recharge events were estimated (Table 4).

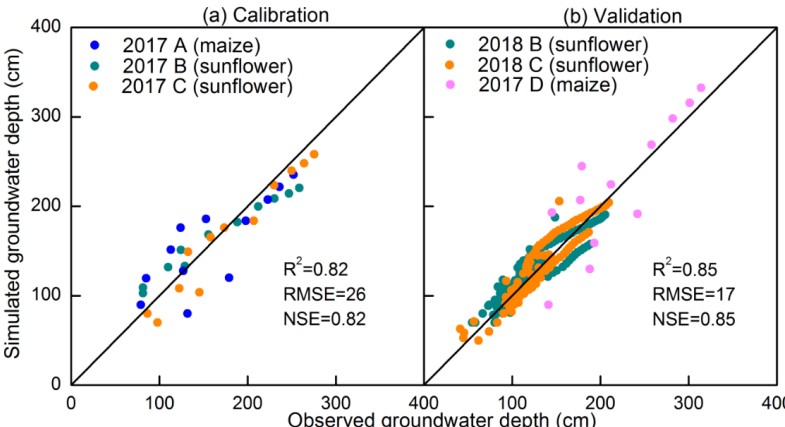


Fig. 15 Comparison of predicted and observed groundwater depth a) calibration and b)
validation.



4.4.3 *Soil salinity*

The only parameter that could be adjusted each year for calibration of the salt

concentrations was the initial salt concentration. Despite the limited calibration, the
observed and predicted values were in close agreement (Fig. 6, 16) with predicted salt
concentrations in the top layers decreasing after an irrigation event as observed. The
Nash Sutcliffe efficiency values were poor (Table 4), likely because the concentration
varied only slightly, and the mean was not predicted accurately. Similarly to the moisture
contents, the salt concentration in the layers below 40 cm was predicted more accurately
than the layers above the 40 cm. Overall, the model can predict the law of salt
concentration fluctuation during crop growth period and the prediction results are
acceptable.

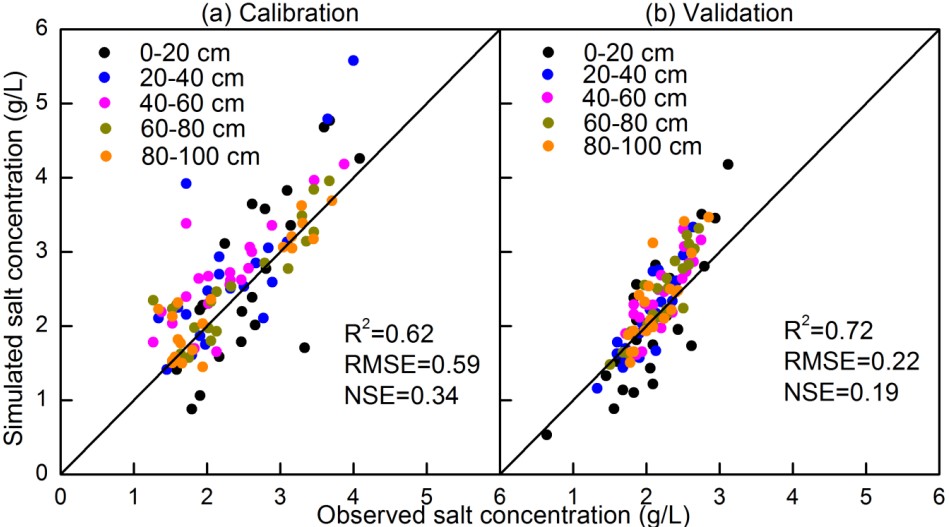


Fig. 16 Comparison of predicted and observed salt concentration during calibration (a)
and validation (b)
**5. Discussion**





The EPICS model is a surrogate model that can be applied in areas with shallow
groundwater. It can simulate the soil moisture content and salt concentration for layers in
the soil, the groundwater depth, upward movement of water from groundwater,
evapotranspiration, and plant growth.
The model is different from traditional models that are based on Richards equation;
instead of calculating the fluxes first, in the EPICS model, the groundwater depth is
calculated first based either on the amount of water removed by evapotranspiration on
days without rain or irrigation or recharge to groundwater on the other days.
Subsequently, when the groundwater is sufficiently shallow and the potential upward flux
from the groundwater is greater than the evaporative demand, the moisture contents are
adjusted so that that soil moisture and groundwater depth are in equilibrium (i.e., field
capacity). In this case, the matric potential is equal to the height above the water table
and the moisture contents can be found with the soil characteristic curve. When the
upward flux is less than the evaporative demand of the atmosphere and crop, the
difference between the upward moisture content is determined by first decreasing the
moisture content below the field capacity. The flux of water in the soil is then calculated
based on the changes in water content. The advantage is fewer input parameters needed
when compared with other numerical models (Šimůnek et al., 1996; Dam et al., 1997).
For example, the hydraulic conductivity is not used in EPICS.
Although the uncertainties of field experimental observations and input data of the
model affected the accuracy of simulation results, EPICS compares well with other models.
Xu et al. (2015) tested the SWAP-EPIC for two lysimeters grown with maize on the same





experimental farm in the Hetao irrigation district where our experiment was carried out.
The SWAP model solves the Richards' Equation numerically with an implicit backward
scheme and is combined by Xu et al. (2015) with the EPIC model. The accuracy of our
simulation results, despite the difference in complexity, are very similar. The moisture
contents were simulated slightly better with EPICS, the groundwater depth was nearly
the same, and the LAI values were predicted more accurately in the SWAP-EPIC model.
Xue et al. (2015) did not simulate the salt content of the soil. Compared to less data and
computational intensive models that are applied in the Yellow River, the soil moisture
content were simulated more accurately by EPICS than in the North China Plain with 30
m deep groundwater by surrogate models of Kendy et al. (2003) and Yang et al. (2015
a,b) and in the Hetao irrigation district by Gao et al. (2017b) and Xue et al. (2018) during
the crop growth period.

To obtain more accurate results in the future, the upward capillary flux from

groundwater needs to be improved. In addition, the evapotranspiration measured
independently, using Eddy covariance (Zhang et al., 2012; Armstrong et al., 2008) and
Bowen ratio-energy balance method (Zhang et al., 2007) should be further used to test
performance of the model in the future study.

The limitation of the EPICS model is it can only be applied in areas where

groundwater is generally less than 3.3 m deep. When the groundwater is deeper than 3.3
m, the field capacity of the surface soil is determined by the moisture content when the
hydraulic conductivity becomes limiting and not by the depth of the groundwater.

Overall, the present model has the advantage that it greatly simplifies the calculation

of the moisture content, groundwater depth and salt content and despite that, gives
results similar to or better than other models applied in the Yellow river basin.
**6. Conclusions**
A novel surrogate field hydrological model called *Evaluation of the Performance of*
*Irrigated Crops and Soils* (EPICS) was developed for irrigated areas with shallow
groundwater. The model was tested with two years experimental data collected by us for
sunflower and one year of maize on replicated fields in the Hetao irrigation district, a
typical arid to semi-arid irrigation district with a shallow aquifer. The EPICS model uses
the soil moisture characteristic curve, upward capillary flux, and groundwater depth to
derive the drainable porosity and predict the soil moisture contents and salinity. The
evaporative flux is calculated with equations in EPIC (Environmental Policy Integrated
Climate) and root distribution equation.
The simulation results show that the EPICS model can predict the soil moisture
content and salt concentration in different soil layers, groundwater depth, and crop
growth on a daily time step with acceptable accuracy during calibration and validation.
The saturated soil moisture content is the most sensitive parameter for soil moisture
content, salt concentration, and ET in our model.
In the future, the model should be tested in other areas with shallow groundwater
that can be found in surface irrigated sites and in humid climates in river plains. Once
fully tested, the EPICS model can be used for optimizing water use at the local scale but,
more importantly, on a watershed scale in closed basins where every drop of water
counts.




**Data availability:** The observed data used in this study are not publicly accessible. These
data have been collected by personnel of the College of Water Resources and Civil
Engineering, China Agricultural University, with funds from various cooperative sources.
Anyone who would like to use these data, should contact Zhongyi Liu, Xianghao Wang
and Zailin Huo to obtain permission.
**Author contributions:** LZ and XW collected the data. ZL, ZH, CW, GH, XX and TS
contributed to the development of the model. The simulations with the model were done
by ZL, ZH and TS. Preparation and revision of the paper were done by ZL under the
supervision of TS and ZH.
**Competing interests:** The authors declare that they have no conflicts of interest.
**Acknowledgements:** Peggy Stevens helped greatly with polishing the English. We thank
Xianghao Wang and Limin Zhang and the technicians in the Shahaoqu experimental
station who helped in collecting data.
**Financial support:** This study has been supported by National Key Research and
Development Program of China (2017YFC0403301) and the National Natural Science
Foundation of China (No. 51639009, 51679236).

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
