# Peer review of "A FIELD VALIDATED SURROGATE MODEL FOR OPTIMUM PERFORMANCE OF IRRIGATED CROPS IN REGIONS WITH SHALLOW SALTY GROUNDWATER"

_Hydrology and Earth System Sciences, 2019_

## Referee Comment (RC1) · Anonymous Referee #1 · 17 Feb 2020

The manuscript presents a field validated surrogate model to simulated various hydro-agricultural variables in an irrigation field with shallow salty groundwater.

The subject is relevant, and the analysis is interesting, however, the manuscript has a sequence of structural flaws which needs to be addressed. In particular:

1. The title mentions the optimum performance of irrigated crop. Optimization is however NOT a topic covered by the analysis, and optimum crop performances are neither reached nor explored. I agree that the simulation model can support irrigation management, and I suggest to re-phrase the title accordingly.

2. Overall, the authors present too much information about the important role of irriga-

tion, and too little and confused regarding the tradeoff between irrigation and salinity.

3. More information on the current status of surrogate modelling in shallow aquifers is needed since it is not clear how the proposed approach contributes with respect to the current status.

4. The methodology is quite clear and thorough, even though it can be lighter if some textbook material is simplified and properly referred to.

5. It would be interesting to present, at the beginning of the methodology, a methodological framework which includes all the experimental steps and summarizes the field and modelling effort, highlighting the interdependences between the two components.

6. The results could be structured differently (some simulation results appear to be presented beforehand)

Minor comments: L59: Add this information in a separate sentence, providing context on the total extension of the basin.

L 97: I recognize that the objective here is to introduce the need for more surrogate models for irrigation areas with shallow aquifers. However, this sentence appears not connected with what stated before.

L 98-104: I believe the flow of thoughts here should be: 1- There are limited modelling resources when GW is near the surface. 2- Shallow aquifers areas are in fact different from their physical. characterization perspective (i.e. explain better lines 94-104). 3- If any modelling has been performed, it is necessary to provide some context (what did Xue et al., 2018; Gao et al., 2017; Liu et al., 2019 do? what were the shortcomings of their modelling experience?). How the current manuscript contributes towards implementing a more reliable-simple-tailored model in the specific application?

L189: Not clear. Do you mean: j is the exogenous variable on which the term before the parenthesis depends?

L 339: Groundwater?

L 466: I would specify that the SA used in this experiment is a qualitative one

L467: outputs?

L472: I wonder if experimental data should be presented in the case-study characterization, and not in the result section.

L473: calibration and validation results

L595: There is no red line

L626: However, information on calibrated and simulated trajectories of those variables are already shown (see for example fig 7). I would re-name the current section or (even better), restructure the results to complement the above discussion with error statistics.

---

## Referee Comment (RC2) · Anonymous Referee #2 · 20 Feb 2020

This manuscript describes a simplified surrogate model for soil water and salinity dynamics in the vadose zone above shallow water table. The model was calibrated using two years of field data in four fields with maize and sunflowers. Overall, the novel approach employed to simplify the input parameters is interesting, and the quality of writing and presentation in the manuscript is nearly prepared for publication. However, the authors should first address a few issues, detailed in the comments below.

General comments

1. Because the manuscript considers shallow groundwater and surface irrigation, it would help the reader to clarify what is meant by "surface irrigation" (which I assume to

indicate flood type irrigation) and differentiate this from irrigation that is supplied from surface water (as opposed to groundwater supplied irrigation).

2. In the introduction, the authors discuss basic soil physics of hydraulic flow under conditions which require considering matric potential. This is one of the primary contributions of this model and analysis. Regarding the potential for the model to inform irrigation optimization, additional review of salinity management in surface irrigated systems would be appropriate. A model of hydraulic flow and salinity is interesting and potentially very helpful, but should be posed in terms of operational considerations that are relevant to irrigators.

3. The authors employed five standard statistical measures of model performance (RMSE. MRE, Nash-Sutcliffe, $R^2$, and regression coefficient). It would add to the mansucript to discuss why these particular measures reflect model performance, or how they complement each other in evaluating the robustness and representativeness of the model outputs. It would strengthen the results if the analysis included some hypothesis testing, beyond the validation and sensitivity analysis which are presented. There is certainly sufficient sampling (both experimental and modelled) to prepare a compelling significance test.

4. The authors refer to "soil moisture content" and "soil water potential" somewhat interchangeably. Here moisture content is referred to at -33kPa, which is a potential. Please be careful is clarifying this distinction here and throughout the manuscript. From line 248, the Brooks and Corey characteristic curve was used to relate soil moisture content to matric potential. The explicit treatment of the water and salinity flux in section 2.3.2 is helpful, but not sufficient, for me or the average reader to keep track of which parameters are modelled explicitly and which are derived, so keeping the units and variables clear will help a great deal.

5. In Figure 9, it is hard to relate how the predicted soil characteristic curve has been fitted to the observed data. The explanation about points being located to the left of

the curve due to mismatched rates of recharge and root extraction makes sense, but not if virtually all the observations do not fall on or near the curve. This becomes more important in the next section on sensitivity analysis. If the sensitive input parameters are, as the authors say, related primarily to soil hydraulic properties, then it would help the reader to understand how the authors addressed uncertainty in these parameters. It is clear that the authors have done substantial work to calibrate the model to

Minor comments

Line 168: I generally agree with the statement that "Finer resolution is not needed for managing water and salt content for irrigation". However, other aspects of irrigation management are managed on shorter time periods, and consider environmental variables that are not well represented by daily averages. I think it is important to specify the limits of any model, especially surrogate models and models that couple processes that operate over different time and spatial scales. As noted by the authors (line 89), surrogate models are not as versatile as complex models. In keeping eith the intent of making the model generally useful under real world conditions, please be more explicit about the range of conditions under which this model has been shown to work.

Line 206: Considering that maize and sunflowers have very different responses to drought and salinity, and different root development/depth, please discuss further why the same ïAd' values were used for both crops. Otherwise the discussion of root functions is adequate for this presentation.

Line 393: It is a minor point, since Figure 3 is not used except to provide a general visual reference, but please check your citation of the GE imagery. In general, include date of the image, and the date that the image was downloaded.

Lines 411-421: Were manual measurements of soil moisture taken in field B at any point in 2018 to calibrate/corroborate the Hydra Probe sensor measurements?

Line 437 and elsewhere: Some symbols (such as theta for volumetric water content)

[Figure]

are italicized at some points in the text, but not in other places or in tables. Please use a consistent symbol and font so that notation is clear throughout the manuscript, and define each symbol at first usage. Also, please be consistent with notation subscripts (f.c., 33, 15, etc.). Because the manuscript describes calculation steps and several equations and several cases for each equation, a table with all notation for variables and subscripts may be very helpful to the reader. Also, I could not find the first usage of ms cmˆ-1, which may need to be explained as millisiemens per centimeter, and is typically noted as mS cmˆ-1, with siemens capitalized.

Line 485: While the period in 2017 is five weeks longer than the period in 2018, this is still a remarkably large difference in reference ET between the two growing seasons. Please offer some explanation why ETref would differ so much in 2017 and 2018.

Lines 543-546: Even if there is not proper documentation, is there some supporting information to corroborate the suspected spillover event? Why would this event occur five times (twice in 2017 and three times in 2018) in field C (located in the center of the other fields) and not be observed in the adjacent fields?

Figures 11-13: Enclose the legend within a box so that legend entries can't be mistaken for data. This is especially evident in figure 11, where the legend entry is the same size as the data.

Section 4.4.3: Apart from a visual resemblance of correlation between the predicted and observed salinity in figure 16, the Rˆ2 and NSE do not support the idea that "the observed and predicted values were in close agreement (Fig. 6, 16 ", nor the claim that " the model can predict the law of salt concentration fluctuation during crop growth period and the prediction results are acceptable." Basedo n my reading, it might be more accurate to say that variability was low on a daily time step, and that initial salt concentration is the most important parameter to measure on a seasonal basis. However, I can see how this would undermine the usefulness of the model, and I do think the model and this work merit further attention. The authors have attributed a potential cause of

low NSE to the low variability, but this is also related to the relatively small sample size. It might help to assess this with some kind of significance test. Here is one possible approach which I found with a quick google scholar search: Ritter et al., 2013, "Performance evaluation of hydrological models: Statistical significance for reducing subjectivity in goodness-of-fit assessments"https://doi.org/10.1016/j.jhydrol.2012.12.004

---

## Author Response (AR1)

**Revision Notes (HESS2019656)**

Dear professor Alberto Guadagnini,

Thank you for allowing us to resubmit a substantially changed manuscript hess-2019-656 entitled "A field validated surrogate model for optimum performance of irrigated crops in regions with shallow salty groundwater". Your evaluation of the manuscript was as follows:

> "On the basis of the comments from the Reviewers, a series of major concerns have emerged. From the totality of these, and considering the Authors' responses, I am willing to give the Authors the opportunity to submit a revised work. It should be clear that it is not my intention to discount any of the comments raised by the reviewers and that, in case the Reviewers are not fully satisfied, the manuscript will be unambiguously released."

Below we have addressed all the helpful comments of the reviewers point by point. These responses reflect the substantive changes in the manuscript to clarify our ideas. In our response and in the revised manuscript we have shown in blue the changed text. For clarity we have not marked the deleted text. We answered each comment in full. This means that if the comment was similar for two reviewers, we repeated some of the earlier text. It should make it easier for the reviewers to check our revisions.

We would like to thank the reviewers and you for the thoughtful comments and for your time. We are looking forward to hearing your evaluation and whether more changes are needed.

With high regard
Zailin Huo, Tammo Steenhuis and Zhongyi Liu.

We would like to thank reviewer 1 for his extensive and thoughtful comments. In this document we give a detailed response to all comments. Below we cite first the comment, this is followed by our response and often by a section how the text will be revised in the manuscript. The text in blue are changes and additions in the original text. For clarity we do not show any of the removed text.

Thank you so much.

Zailin, Tammo and Zhongyi

**Major comments:**

**Comment** 1. The title mentions the optimum performance of irrigated crop. Optimization is how- ever NOT a topic covered by the analysis, and optimum crop performances are neither reached nor explored. I agree that the simulation model can support irrigation management, and I suggest to re-phrase the title accordingly.

**Response:** Thanks for your suggestion. We agree that the title of the manuscript does not represents its content. In the revised manuscript, the title was changed as:

> "A FIELD VALIDATED SURROGATE CROP MODEL FOR PREDICTING ROOTZONE MOISTURE AND SALT CONTENT IN REGIONS WITH SHALLOW GROUNDWATER"

**Comment** 2. Overall, the authors present too much information about the important role of irrigation, and too little and confused regarding the tradeoff between irrigation and salinity.

**Response:** Thanks for your suggestion. As we know, irrigation practices are main method to leach salt and weaken the influence for irrigated agriculture, and many researchers analyzed the tradeoff between irrigation and soil salinity (Letey et al., 2011; Hanson et al., 2oo8; Pereira et al., 2002; Minhas et al., 2020). In the section 4.1.4, we analyzed

> "… The soil salinity concentration was decreasing during an irrigation event due to dilution and then gradually increasing partly due to evaporation of the water. Some of the soil salt was transported to the layers below during irrigation and some salt was moving upward with the evaporation from the surface. As expected,

after the harvest, the autumn irrigation decreased the salt concentration from fall 2017 to spring 2018."

The detailed mechanism between irrigation and soil salinity was not explored in this manuscript. Therefore, much more information about the tradeoff between irrigation and soil salinity was not analyzed. We add some studies about the tradeoff between irrigation and soil salinity in the introduction section of the revised manuscript as follows:

"… However, at the same time, capillary upward moving water carries salt from the groundwater increasing the salt in the upper layers of the soil leading to soil degradation and possibly decreasing yields and change of crop patterns to more salt tolerant crops (Guo et al., 2018; Huang et al., 2018). The leaching of salts with irrigation water is necessary and useful for irrigated agriculture (Letey et al., 2011). In north China, the fields are commonly irrigated in the autumn before soil freezing to leach salts and provide water for first growth after deeding in the following year (Feng et al., 2005; Pereira et al., 2007).

Tradeoffs between irrigation practices and soil salinity were studied by a lot of researchers (Hanson et al., 2008; Pereira et al., 2002, 2009; Minhas et al., 2020). Minhas et al. (2020) give a brief review of crop evapotranspiration and water management issues when coping with salinity in irrigated agriculture. Phogat et al. (2020) assessed the effects of long-term irrigation on salt build-up in the soil under unheated greenhouse conditions by the UNSA-TCHEM and HYDRUS-1D (Phogat et al., 2020)."

**Comment** 3. More information on the current status of surrogate modelling in shallow aquifers is needed since it is not clear how the proposed approach contributes with respect to the current status.

**Response:** Thanks for your suggestion. Actually studies about the surrogate model in shallow aquifers are relatively rare compared with studies in deep groundwater depth. Here we analyzed the necessary of building surrogate models for areas with shallow aquifer.

"Simple surrogate models are abundant in China for areas where the groundwater is deeper than approximately 10 m (Kendy et al., 2003; Chen et al., 2010; Ma et al., 2013; Li et al., 2017; Wu et al., 2016), but are limited and relatively scarce for areas where the groundwater is near the surface in the arid to semi-arid areas (Xue et al., 2018; Gao et al., 2017; Liu et al., 2019). In these areas with shallow

aquifer, the upward groundwater flux from groundwater is an important factor in meeting the evapotranspiration demand of the crop (Babajimopoulos et al., 2007; Yeh and Famiglietti, 2009). The advantage of applying surrogate models in areas with shallow aquifer is that they can simulate the hydrological process with fewer parameters using simpler and computationally less demanding mathematical relationships than the traditional finite element or difference models (Wu et al., 2016; Razavi et al., 2012)."

**Comment** 4. The methodology is quite clear and thorough, even though it can be lighter if some textbook material is simplified and properly referred to.

**Response:** We are aware that the text is pretty basic. However, soil physics is not being taught in many universities especially in the USA and we prefer therefore to explain it well so that a wider audience might understand why shallow groundwater can modeled with considering the conductivity.

**Comment** 5. It would be interesting to present, at the beginning of the methodology, a methodological framework which includes all the experimental steps and summarizes the field and modelling effort, highlighting the interdependences between the two components.

**Response:** Thank you for your suggestion. The experimental steps are discussed in the section after the model description. We added the following in the last paragraph of introduction section of the revised manuscript:

"In the following section we present first the theoretical background of the surrogate model. The model consists of crop growth module and a vadose zone module. This is followed by detailed description of the two-year field experiments staring in 2017 in the Hetao irrigation district where maize and sunflower were irrigated by flooding the field. The experimental results consisting of climate data, irrigation application, crop growth parameters, moisture and salt content and groundwater depth are used to calibrate and validate the model."

**Comment** 6. The results could be structured differently (some simulation results appear to be presented beforehand)

**Response:** We are grateful for your suggestion. In the results section, the experimental data was analyzed first in order to avoid showing the observed

experimental data at the time when it is compared with model simulation results. This is not ideal but we found this the least confusing.

**Minor comments:**

**Comment** 1. L59: Add this information in a separate sentence, providing context on the total extension of the basin.

**Response:** Thanks for your suggestion and we moved this sentence to Section 3.1.

> **"**The groundwater depth is between 0.5-3 m. Regional exchange of groundwater is minimal due to low gradient of 0.01-0.025 (Xu et al., 2010). Thus, the groundwater flows mainly vertically with minimum lateral flow in the regional scale. Over 50% of the total irrigated cropland, 5250 $km^2$ in the Hetao irrigation district in the Yellow River basin, is affected by salinity (Feng et al., 2005).**"**

**Comment** 2. L 97: I recognize that the objective here is to introduce the need for more surrogate models for irrigation areas with shallow aquifers. However, this sentence appears not connected with what stated before.

**Response:** Thank you for your suggestion. This sentence is used to stress the importance of matric potential in the area with shallow groundwater. In the revise manuscript, it was revised as

> "The change in matric potential is often ignored in these surrogate models for soils with a deep groundwater table. However, for areas with shallow aquifers (i.e., less than approximately 3 m), the matric potential cannot be ignored. The flow of water is upward when the absolute value of matric potential is greater than the groundwater depth or downward when it is less than the groundwater depth (Gardner, 1958; Gardner et al., 1970a; b; Steenhuis et al., 1988). The field capacity in these soils is reached when the hydraulic gradient is constant (i.e., the constant value of sum of matric potential and gravity potential). In this case, the soil water is in equilibrium and no flow occurs.
>
> Xue et al. (2018) and Gao et al. (2017), developed models for the shallow groundwater, but used field capacities and drainable porosities that were calibrated and independent of the depth of the groundwater. This is inexact when the groundwater is close to the surface. Liu et al. (2019), used for simulating shallow groundwater the same type of model as described in this paper but

calibrated crop evaporation and did not simulate the salt concentrations in the soil. This made their model less useful for practical application."

**Comment** 3. L 98-104: I believe the flow of thoughts here should be: 1- There are limited modelling resources when GW is near the surface. 2- Shallow aquifers areas are in fact different from their physical. characterization perspective (i.e. explain better lines 94-104). 3- If any modelling has been performed, it is necessary to provide some context (what did Xue et al., 2018; Gao et al., 2017; Liu et al., 2019 do? what were the shortcomings of their modelling experience?). How the current manuscript contributes towards implementing a more reliable-simple-tailored model in the specific application?

**Response:** Thank you for your suggestion. In the revised manuscript, we added some information in the next paragraph

"The change in matric potential is often ignored in these surrogate models for soils with a deep groundwater table. However, for areas with shallow aquifers (i.e., less than approximately 3 m), the matric potential cannot be ignored. The flow of water is upward when the absolute value of matric potential is greater than the groundwater depth or downward when it is less than the groundwater depth (Gardner, 1958; Gardner et al., 1970a; b; Steenhuis et al., 1988). The field capacity in these soils is reached when the hydraulic gradient is constant (i.e., the constant value of sum of matric potential and gravity potential). In this case, the soil water is in equilibrium and no flow occurs.

Xue et al. (2018) and Gao et al. (2017), developed models for the shallow groundwater, but used field capacities and drainable porosities that were calibrated and independent of the depth of the groundwater. This is inexact when the groundwater is close to the surface. Liu et al. (2019), used for simulating shallow groundwater the same type of model as described in this paper but calibrated crop evaporation and did not simulate the salt concentrations in the soil. This made their model less useful for practical application.

Because of the shortcomings in the above complex models, we avoided the use of a constant drainable porosity and considered the crop growth and thus improved the surrogate model in our last study (Liu et al., 2019). The objective of this research was to develop a field validated surrogate model that could be used to simulate the water and salt movement and crop growth in irrigated areas with shallow groundwater and salinized soil with a minimum of input parameters. To validate the surrogate model, we performed a 2-year field experiment in the

Hetao irrigation district that investigated the change in soil salinity, moisture content, groundwater depth and maize and sunflower growth during the growing season."

**Comment** 4. L189: Not clear. Do you mean: j is the exogenous variable on which the term before the parenthesis depends?

**Response:** Apologies for the unclear expression. In this study, j is the number of soil layer and t is the day number. We add this information in the revised manuscript

"where j is the number of soil layer and t is the day number, $T_p(t)$ is the total potential transpiration….."

**Comment** 5. L 339: Groundwater?

**Response:** It is "water". Here we tried to introduce the movement of soil water and groundwater, not just groundwater.

**Comment** 6. L 466: I would specify that the SA used in this experiment is a qualitative one.

**Response:** Yes, as this reviewer point out, this simple parameter sensitivity analysis method only produces the qualitative results to show which parameters are important to output of the model. This is useful to determine related parameters to use the model. We have explained these in the 3.4 section.

**Comment** 7. L467: outputs?

**Response:** Apologies for this vague expression. It was revised as

"Each parameter was varied over a range of -30% to 30% to derive the corresponding impact on the model output of soil moisture, groundwater depth, soil salinity, leaf area index and actual evapotranspiration."

**Comment** 8. L472: I wonder if experimental data should be presented in the case-study characterization, and not in the result section.

**Response:** Thanks for your suggestion. It is always difficult to decide how to structure a paper. The field experiment was carried out by us and therefore we believe that it should be in the results section. If the experiment was not carried out by the authors, it should certainly be in the case study characterization.

**Comment** 9. L473: calibration and validation results

**Response:** Thanks for your suggestion. It was revised as

"The 2017 and 2018 experimental data of the Shahaoqu farmers' fields in the Hetao irrigation district (Fig.3) are presented first, followed by the calibration and validation results of the CROP and VADOSE modules of EPICS model."

**Comment** 10. L595: There is no red line

**Response:** Apologies for the mistake. It was revised as

"The pink line is the fit with the Brooks-Corey equation."

**Comment** 11. L626: However, information on calibrated and simulated trajectories of those variables are already shown (see for example fig 7). I would re-name the current section or (even better), restructure the results to complement the above discussion with error statistics.

**Response:** We are grateful for your suggestion. The simulation results were shown with the experimental results because we analyzed the experimental data first. And this section is about the comparison of simulation results and experimental results and the model results error analysis. It was revised as "4.4 Model calibration and validation with field data".

References

[revised manuscript text omitted]

Yeh, Pat J-F., Famiglietti, J.: Regional Groundwater Evapotranspiration in Illinois. J. Hydrometeorol., 10:464-478. https:// doi.org/10.1175/2008JHM1018.1. 2009

**Responses to the comments of Reviewer #2:**

We would like to thank reviewer 2 for his extensive and thoughtful comments. In this document we give a detailed response to all comments. Below we cite first the comment, this is followed by our response and often by a section how the text will be revised in the manuscript. The text in blue are changes and additions in the original text. For clarity we do not show any of the removed text.

Thank you so much.

Zailin, Tammo and Zhongyi

**General comments:**

**Comment** 1. Because the manuscript considers shallow groundwater and surface irrigation, it would help the reader to clarify what is meant by "surface irrigation" (which I assume to indicate flood type irrigation) and differentiate this from irrigation that is supplied from surface water (as opposed to groundwater supplied irrigation).

**Response:** Thanks for your suggestion. It was revised as

"In arid and semi-arid surface irrigation districts with flood irrigation and without a drainage infrastructure, the groundwater table is close to the surface because more water has been applied than crop evapotranspiration."

Actually, in the section 3.2, we explained it is flood irrigation "The fields were irrigated by flooding the field ranging from two to five times during the growing season (Table 1)."

**Comment** 2. In the introduction, the authors discuss basic soil physics of hydraulic flow under conditions which require considering matric potential. This is one of the primary contributions of this model and analysis. Regarding the potential for the model to inform irrigation optimization, additional review of salinity management in surface irrigated systems would be appropriate. A model of hydraulic flow and salinity is interesting and potentially very helpful, but should be posed in terms of operational considerations that are relevant to irrigators.

**Response:** We are grateful for your suggestion. The model can potentially be used to optimum water use efficiency and crop yield but this was not explored in this manuscript. Therefore, the title of the manuscript was revised as

"A FIELD VALIDATED SURROGATE CROP MODEL FOR PREDICTING ROOTZONE MOISTURE AND SALT CONTENT IN REGIONS WITH SHALLOW GROUNDWATER"

Besides, the last three paragraphs in the introduction part was revised as

"The change in matric potential is often ignored in these surrogate models for soils with a deep groundwater table. However, for areas with shallow aquifers (i.e., less than approximately 3 m), the matric potential cannot be ignored. The flow of water is upward when the absolute value of matric potential is greater than the groundwater depth or downward when it is less than the groundwater depth (Gardner, 1958; Gardner et al., 1970a; b; Steenhuis et al., 1988). The field capacity in these soils is reached when the hydraulic gradient is constant (i.e., the constant value of sum of matric potential and gravity potential). In this case, the soil water is in equilibrium and no flow occurs.

Xue et al. (2018) and Gao et al. (2017), developed models for the shallow groundwater, but used field capacities and drainable porosities that were calibrated and independent of the depth of the groundwater. This is inexact when the groundwater is close to the surface. Liu et al. (2019), used for simulating shallow groundwater the same type of model as described in this pater but calibrated crop evaporation and did not simulate the salt concentrations in the soil. This made their model less useful for practical application.

Because of the shortcomings in the above complex models, we avoided the use of a constant drainable porosity and considered the crop growth and thus improved the surrogate model in our last study (Liu et al., 2019). The objective of this research was to develop a field validated surrogate model that could be used to simulate the water and salt movement and crop growth in irrigated areas with shallow groundwater and salinized soil with a minimum of input parameters. To validate the surrogate model, we performed a 2-year field experiment in the Hetao irrigation district that investigated the change in soil salinity, moisture content, groundwater depth and maize and sunflower growth during the growing season."

**Comment** 3. The authors employed five standard statistical measures of model performance (RMSE. MRE, Nash-Sutcliffe, $R^2$, and regression coefficient). It would add to the manuscript to discuss why these particular measures reflect model performance, or how they complement each other in evaluating the robustness and representativeness of the model outputs. It would strengthen the results if the analysis

included some hypothesis testing, beyond the validation and sensitivity analysis which are presented. There is certainly sufficient sampling (both experimental and modelled) to prepare a compelling significance test.

**Response:** Thanks for your suggestion. When only one indicator was used to quantify the goodness-of-fit of observations against model simulated values may lead to incorrect verification of the model (Ritter and Muñoz-Carpena, 2013). The combination of these statistical indicators were used to quantify the performance of model. In this manuscript, five indicators which were widely used to evaluate the performance of hydrological models were used (Ren et al. 2016; Xu et al., 2016). In the revised manuscript, we explained these parameters:

"…where $N$ is the total number of observations; $P_i$ and $O_i$ are the $i^{th}$ model predicted and observed values ($i$=1,2,3…N), respectively; $\bar{O}$ and $\bar{P}$ are the mean observed values and predicted values, respectively. The RMSE is used to evaluate the bias of the measured data and predicted data. The MRE can evaluate the credibility of the measured data. The NSE is usually used to evaluate the quality of the hydrological models. The R2 is used to measure the fraction of the dependent variable total variation that can be explained by the independent variable. And the regression coefficient represents the influence of the independent variable on the dependent variable in the regression equation. The value of RMSE and MRE close to 0 indicates good model performance. The value of NSE ranges from -∞ to 1. NSE=1 means a perfect fit while the negative NSE value indicates the mean observed value is a better predictor than the simulated value (Moriasi et al., 2007). For b and $R^2$, the value closest to 1 indicates good model predictions."

**Comment** 4. The authors refer to "soil moisture content" and "soil water potential" somewhat interchangeably. Here moisture content is referred to at -33kPa, which is a potential. Please be careful is clarifying this distinction here and throughout the manuscript. From line 248, the Brooks and Corey characteristic curve was used to relate soil moisture content to matric potential. The explicit treatment of the water and salinity flux in section 2.3.2 is helpful, but not sufficient, for me or the average reader to keep track of which parameters are modelled explicitly and which are derived, so keeping the units and variables clear will help a great deal.

**Response:** We are grateful for your suggestion. In our manuscript, we assume that the field capacity is not constant but a unique relationship as function of the matric potential. The matric potential is equal to the height above the water table (when

expressed as a suction in length units).

The reason is that we refer to the moisture content at –33KPa that we do not use the traditional definition of field capacity for soil with at water table below 3.3 m when the conductivity becomes limiting. So we need to call the traditional field capacity something different, (i.e., moisture content at -33Kpa or 0.33 bar) the matric potential is -33kpa, which can also be expressed as the distance of the point above the water table. In the revised manuscript, we improved the readability of the section 2.3.2

[revised manuscript text omitted]

**Comment 5.** In Figure 9, it is hard to relate how the predicted soil characteristic curve has been fitted to the observed data. The explanation about points being located to the left of the curve due to mismatched rates of recharge and root extraction makes sense, but not if virtually all the observations do not fall on or near the curve. This becomes more important in the next section on sensitivity analysis. If the sensitive input parameters are, as the authors say, related primarily to soil hydraulic properties, then it would help the reader to understand how the authors addressed uncertainty in these parameters. It is clear that the authors have done substantial work to calibrate the model to

**Response:** We agree that we could have explained this section better: The points that are to the left of the curve are those when the evaporative demand is greater than the upward flux. At these moisture content the soil moisture content is less that the traditional definition of field capacity at -33 KPa and hence the Darcy flux is extremely small and the soil dries out due to water uptake by the roots. Therefore, we omit these points when fitting the soil characteristic curve parameters (the bubbling pressure, the saturated moisture content and the exponent). Only the saturated moisture content is a very sensitive parameter (see section 4.3 Parameter uncertainty). There is little uncertainty in defining the saturated moisture content. There is a certain

art in finding the other two parameters, but they are not very sensitive luckily in the final model outcome. (See section 4.3 on the sensitivity analysis). In performing the next experiment, a tensiometer should be installed in the soil and measure the suction and moisture content at the same time). This would avoid the current uncertainty.

We changed the text in the paragraph as follows:

> "To simulate the soil moisture content and to derive drainable porosity as a function of water table depth, the soil moisture characteristic curves were derived by plotting the observed soil moisture content in 2017 and 2018 versus the height above the water table to the soil surface for the five soil layers in Fig. 9. The Brooks-Corey equation (Brooks and Corey, 1964) was fitted through outer envelope of the points. The parameters of the Brooks-Corey equation were adjusted through a trial and error to obtain the best fit (Table 3a). In Fig. 9, points on the left side of the soil moisture characteristic curve (moisture content smaller than the field capacity) were due to water removal at times when evaporative demand was greater than the upward water flux. Under these conditions the conductivity is limiting in the soil and there is no relationship between groundwater depth and matric potential. Since we take the water table depth as proxy for matric potential, these points are omitted when drawing the soil characteristic curve. The few points at the right of the soil moisture characteristic curve indicate the soil moisture was greater than field capacity and matric potential and groundwater were not yet at equilibrium after an irrigation event."

**Minor comments:**

**Comment** 1. Line 168: I generally agree with the statement that "Finer resolution is not needed for managing water and salt content for irrigation". However, other aspects of irrigation management are managed on shorter time periods and consider environmental variables that are not well represented by daily averages. I think it is important to specify the limits of any model, especially surrogate models and models that couple processes that operate over different time and spatial scales. As noted by the authors (line 89), surrogate models are not as versatile as complex models. In keeping eiththe intent of making the model generally useful under real world

conditions, please be more explicit about the range of conditions under which this model has been shown to work.

**Response:** Thank you for your suggestion. Here, we aimed to stress our study is daily time step and cannot be used to simulate the instantaneous change of water and salt. Furthermore, in the revised manuscript, the section 2.3, it was revised as:

> "In the next section, the equations of the CROP in the VADOSE modules are presented. The calculations are carried out sequentially on a daily time step. This model predicts field daily soil water, salt content and crop growth, which are critical parameters for irrigation water management. For field and regional water management and irrigation policy development, resolution of daily time step is sufficient. Finer resolution is not needed for managing water and salt content for irrigation…"

**Comment** 2. Line 206: Considering that maize and sunflowers have very different responses to drought and salinity, and different root development/depth, please discuss further why the same $\delta$ values were used for both crops. Otherwise the discussion of root functions is adequate for this presentation.

**Response:** We tried to find a reference for "$\delta$" of sunflower, but we did not come across a reference. The study of Chen et al. (2019) also used the same $\delta$ for all crops in his model. For both sunflower and maize the main roots are both in the upper 90cm. Thus we use the same $\delta$ here.

**Comment** 3. Line 393: It is a minor point, since Figure 3 is not used except to provide a general visual reference, but please check your citation of the GE imagery. In general, include date of the image, and the date that the image was downloaded.

**Response:** Thanks for your suggestion. Figure 3 was used to show the geographical location of experimental field in section 3.1. The GE imagery is the day of April 8, 2019 and was downloaded on April 8, 2019. We add this information in the revised manuscript.

[Figure]

Fig. 3 Location of the Shahaoqu experimental field (Note: The figure was downloaded from Google earth. The imagery is taken on April 8, 2019)

**Comment** 4. Lines 411-421: Were manual measurements of soil moisture taken in field B at any point in 2018 to calibrate/corroborate the Hydra Probe sensor measurements?

**Response:** Yes, the soil moisture content of field B was also manual measured to calibrate the Hydra Probe sensor. And in the manuscript we only show the calibrated data of the Hydra Probe sensor to compare with the simulation results. In the section 3.2, we add this information:

> "…The soil moisture content for the four experimental fields in 2017 and for field C in 2018 during the crop growing season was measured every 7-10 days at the depths of 0-20, 20-40, 40-60, 60-80, 80-100 cm by taking soil samples and oven drying. In 2018, in addition the soil moisture content at same depths was monitored daily using Hydra Probe Soil Sensors (Stevens Water Monitoring System Inc., Portland, OR, USA) in field B except the oven drying method. The Hydra Probe was calibrated using the intermittent manual measurements. In 2017, the groundwater depths were manually measured in all four experimental fields about every 7-10 days…."

**Comment** 5. Line 437 and elsewhere: Some symbols (such as theta for volumetric water content) are italicized at some points in the text, but not in other places or in

tables. Please use a consistent symbol and font so that notation is clear throughout the manuscript, and define each symbol at first usage. Also, please be consistent with notation subscripts (f.c., 33, 15, etc.). Because the manuscript describes calculation steps and several equations and several cases for each equation, a table with all notation for variables and subscripts may be very helpful to the reader. Also, I could not find the first usage of ms cmˆ-1, which may need to be explained as millisiemens per centimeter, and is typically noted as mS cmˆ-1, with siemens capitalized.

**Response:** Our apologies for missing this. The first mS cmˆ-1 appeared in Eq. 6 which used to calculate the salt stress coefficient.

We revised the notations to keep it consistent and a table with all notations for variables and subscripts was added in the revised manuscript.

| Nomenclature | | | |
|---|---|---|---|
| $ET_0$ | Reference evapotranspiration (mm) | $p$ | Fraction of readily avaiable soil water relative to the total avaiable soil water () |
| $ET_P$ | Potential evapotranspiration (mm) | $S$ | Salt stress coefficient () |
| $E_p$ | Potential evaporation (mm) | $B$ | Crop specific parameter (%) |
| $T_p$ | Potential transpiration (mm) | $k_y$ | Factor that affects crop yield () |
| $E_a$ | Actual evporation (mm) | $ECe$ | Electrical conductivity of the soil saturation extract (mS cm$^{-1}$) |
| $T_a$ | Actual transpiration (mm) | $EC_{ethreshold}$ | Threshold of the electrical conductivity of the soil saturation extract when the crop yield becomes affected by salt (mS cm$^{-1}$) |
| | | $EC_{1:5}$ | Electrical conductivity of the soil extract that soil samples mixed with distilled water in a proportion of 1:5 (mS cm$^{-1}$) |
| $K_c$ | Crop coefficient() | | |
| $\tau$ | Development stage of the leaf canopy() | $\theta_s$ | Soil mositure content at saturation (cm$^{-3}$ cm$^{-3}$) |
| $r_T$ | Root function for transpiration () | $\varphi_b$ | Bubbling pressure (cm) |
| $r_E$ | Root function for transpiration () | $\varphi_m$ | Matric potential (cm) |
| $j$ | Number of soil layer() | $\lambda$ | Pore size distribution index |
| $LAI$ | Leaf area index() | $h$ | Groundwater depth (cm) |
| $T_{mean}$ | Mean daily temperature (℃) | $z$ | Depth of the point below the soil surface (cm) |
| $T_{mx}$ | Maximum daily temperature (℃) | $W_{fc}(h)$ | Total water content at field capacity of the soil profile over a prescribed depth (cm) |

| | | | |
|---|---|---|---|
| $T_{mn}$ | Minimum daily temperature (℃) | $L(j)$ | Height of layer j (cm) |
| $LAI_{mx}$ | Maximum leaf area index | $\mu$ | Drainable porosity |
| $RD_{mx}$ | Maximum root depth (cm) | $P$ | Precipitation (mm) |
| $K_b$ | Dimensionless canopy extinction coefficient | $I$ | Irrigation (mm) |
| $PHU$ | Total potential heat units required for crop maturation (℃) | $n$ | Number of soil layers |
| $Z1j$ | Depth of the upper boundaries of soil layer j (cm) | $R_{gw}$ | Percolation to groundwater (mm) |
| $Z2j$ | Depth of the lower boundaries of the soil layer for $r_E(j,t)$; root depth or the lower boundaries of the soil layer for $r_T(j,t)$ (cm) | $R_w(j\text{-}1,t)$ | Percolation rate to layer j from layer j-1 at day t (mm) |
| $\delta$ | Water use distribution parameter | $C(j,t)$ | Salt concentration of layer j at day t (g $L^{-1}$) |
| $k_E$ | Water stress coefficient for evaporation | $C_I$ | Salt conctration of irrigation water (g $L^{-1}$) |
| $k_T$ | Water stress coefficient for transpiration | $C_{gw}$ | Salt contration of groundwater (g $L^{-1}$) |
| $\theta$ | Soil moisture content ($cm^{-3}$ $cm^{-3}$) | $U_{gw}$ | Actual upward flux of groundwater (mm) |
| $\theta_{fc}$ | Soil moisture content at field capacity ($cm^{-3}$ $cm^{-3}$) | $U_{gw,max}$ | Maximun upward flux of groundwater (mm) |
| $\theta_r$ | Soil moisture content at wilting point ($cm^{-3}$ $cm^{-3}$) | $a$ | Constant used for calcualtion of $U_{gw,max}()$ |
| $f_{shape}$ | Shape factor of $k_T$ curve () | $b$ | Constant used for calcualtion of $U_{gw,max}()$ |

**Comment** 6. Line 485: While the period in 2017 is five weeks longer than the period in 2018, this is still a remarkably large difference in reference ET between the two growing seasons. Please offer some explanation why ETref would differ so much in 2017 and 2018.

**Response:** The Penman-Monteith equation was used to calculate the reference evapotranspiration (Allen, 1998). The total precipitation was 63mm and 108mm in 2017 and 2018 during study period, respectively. There were more rainfall days in 2018 than in 2017, which lead to the total $ET_0$ is greater in 2017 than in 2018 during June 1 to September. Besides, the wind speed was high and the evapotranspiration was high in May. In the study of Ren et al. (2017) and Miao (et al. 2016), the mean daily $ET_0$ is over 6 mm per day. Hence, the $ET_0$ during the study period in 2017 is greater than in 2018. In the revised manuscript, we add this explanation:

"…The reference evapotranspiration ranged from 1 mm $d^{-1}$ to a maximum of 6.4

mm d$^{-1}$ during crop growing period (Fig. 4). The total reference evapotranspiration from May 10 to September 30, 2017 was 595 mm and 368 mm from June 1 to September 15, 2018. The reason was that there were more rainfall days in June, July and September in 2018 than in 2017, which increased the amount of water available for the evapotranspiration by the crop in 2018. In addition, the wind speed was high in May that increase the evapotranspiration was elevated. In the study of Ren et al. (2017) and Miao (et al. 2016), the mean $ET_0$ was over 6 mm per day on May. Hence, the $ET_0$ during the study period in 2017 was greater than in 2018."

**Comment** 7. Lines 543-546: Even if there is not proper documentation, is there some supporting information to corroborate the suspected spillover event? Why would this event occur five times (twice in 2017 and three times in 2018) in field C (located in the center of the other fields) and not be observed in the adjacent fields?

**Response:** We discovered this increase in water table without rainfall or irrigation during testing of the model. It is therefore difficult to reconstruct exactly what happened. The spillover event is just our guess about this strange phenomenon and we have no supporting information to corroborate this suspect. In our last study (Liu et al., 2019), we also have same phenomenon. And our hypotheses of the increase of the groundwater depth due to irrigation in a nearby field is that early in the season the cracks in the structured clays were not fully closed and these could have transported some of the water across the field. It is not something that can be predicted by a standard finite difference or element model since the conductivity is so small for this site. So it is unexpected (or curious).

Another explanation might be that in a nearby field was irrigated increasing the water table. Since pressure travels with the speed of sound and there is only a tiny amount of water displacement necessary to change the water table height when the pressure is changed, this could also cause the water table height increase. Clearly more research is needed to define the cause.

We changed the paragraph to include the above as:

"The variation in groundwater depth during the growing season was very similar for both years and in all fields. The groundwater depth for all fields was between 50 and 100 cm from the surface after an irrigation event and then decreased to around 150 cm before the next irrigation or rainfall (Fig.7). Only after the last irrigation in August 2017 did the water table decrease to below 250 cm and to around 200 cm in 2018. Field D followed the same pattern but the

groundwater was more down from the surface. In several instances, the groundwater table increased without an irrigation or rainfall event in sunflower field C (Fig. 7c and 7e). This was likely related to an irrigation event either from an irrigation in nearby field that affected the overall water table or an accidental irrigation that was not properly documented. We estimated the amount of irrigation water based on the change in moisture content in the soil profile (orange bars in Fig. 7c and 7e). Finally, there was a notable rise in the water table of an mean 375mm "autumn irrigation" after harvest between the end of 2017 (Figs. 7 a, b, c) and the beginning of 2018 (Figs. 7 d, e, f), which is a common practice in the Jiefangzha irrigation district to leach the salt that has accumulated in the profile during the growing periods."

**Comment** 8. Figures 11-13: Enclose the legend within a box so that legend entries can't be mistaken for data. This is especially evident in figure 11, where the legend entry is the same size as the data.

**Response:** Thanks for your suggestion. And the legend was enclosed in a box as follows:

[Figure]

Fig. 11 Comparison of predicted and observed actual evapotranspiration: a) Calibration and b) Validation

[Figure]

Fig.12 Comparison of predicted and observed crop height: a) Calibration and b) Validation

[Figure]

Fig. 13 Comparison of predicted and observed LAI: a) Calibration and b) validation

**Comment** 9. Section 4.4.3: Apart from a visual resemblance of correlation between the predicted and observed salinity in figure 16, the Rˆ2 and NSE do not support the idea that "the observed and predicted values were in close agreement (Fig. 6, 16 ", nor the claim that " the model can predict the law of salt concentration fluctuation during crop growth period and the prediction results are acceptable." Based on my reading, it might be more accurate to say that variability was low on a daily time step, and that initial salt concentration is the most important parameter to measure on a seasonal basis. However, I can see how this would undermine the usefulness of the model, and I do think the model and this work merit further attention. The authors have attributed a potential cause of low NSE to the low variability, but this is also related to the relatively small sample size. It might help to assess this with some kind of significance test. Here is one possible approach which I found with a quick google

scholar search: Ritter et al., 2013, "Performance evaluation of hydrological models: Statistical significance for reducing subjectivity in goodness-of-fit assessments" https://doi.org/10.1016/j.jhydrol.2012.12.004

**Response:** Thank you for providing the reference for the Ritter et al (2013) reference. We agree that overstated the goodness of fit for the salinity concentration in the soil. Visually the fit is reasonable, but the statistics do not show that. Clearly more work is needed, but before it can be modeled, other and detailed information is needed. Especially we the effect of the autumn irrigation after crop harvest and the freezing of soil on salt transport need to be studied Based on the comment above we changed section 4.3.3 as

[revised manuscript text omitted]

---

## Referee Report (RR1)

**Abstract**

Optimum performance of irrigated crops in regions with shallow saline groundwater requires a careful balance between application of irrigation water and upward movement of salinity from the groundwater. Few field validated surrogate models are available to aid in the management of irrigation water under shallow groundwater conditions. The objective of this research is to develop a model that can aid in the management using a minimum of input data that is field validated. In this paper a 2-year field experiment was carried out in the Hetao irrigation district in Inner Mongolia, China and a physically based integrated surrogate model for arid irrigated areas with shallow groundwater was developed and validated with the collected field data. The integrated model that links crop growth with available water and salinity in the vadose zone is called Evaluation of the Performance of Irrigated Crops and Soils (EPICS). EPICS recognizes that field capacity is reached when the matric potential is equal to the height above the groundwater table and thus not by a limiting hydraulic conductivity. In the field experiment, soil moisture contents and soil salt conductivity at 5 depths in the top 100 cm, groundwater depth, crop height, and leaf area index were measured in 2017 and 2018. The field results were used for calibration and validation of EPICS. Simulated and observed data fitted generally well during both calibration and validation. The EPICS model that can predict crop growth, soil water, groundwater depth and soil salinity can aid in optimizing water management in irrigation districts with shallow aquifers.

**Key words:** Surrogate hydrological model, irrigated crops, shallow aquifer

Number: 1  Author:  Subject: Highlight  Date: 2020-06-18 16:41:32

In light of the many uses for the word performance, perhaps "optimum managaement of irrigated crops" would be more straightforward.

Nomenclature

$ET_0$      Reference evapotranspiration (mm)

$ET_P$      Potential evapotranspiration (mm)

$E_p$      Potential evaporation (mm)

$T_p$      Potential transpiration (mm)

$E_a$      Actual evporation (mm)

$T_a$      Actual transpiration (mm)

$K_c$      Crop coefficient()

$\tau$      Development stage of the leaf canopy()

$r_T$      Root function for transpiration ()

$r_E$      Root function for transpiration ()

$j$      Number of soil layer()

LAI      Leaf area index()

$T_{mean}$      Mean daily temperature (℃)

$T_{mx}$      Maximum daily temperature (℃)

$T_{mn}$      Minimum daily temperature (℃)

$LAI_{mx}$      Maximum leaf area index

$RD_{mx}$      Maximum root depth (cm)

$K_b$      Dimensionless canopy extinction coefficient

PHU      Total potential heat units required for crop maturation (℃)

$Z_{1j}$      Depth of the upper boundaries of soil layer j (cm)

[revised manuscript text omitted]

---

## Author Response (AR2)

**Revision Notes (HESS2019656)**

July 10, 2020

Dear Professor Alberto Guadagnini,

Thank you for allowing us to resubmit the minor changed manuscript Hess-2019-656 entitled "A FIELD VALIDATED SURROGATE CROP MODEL FOR PREDICTING ROOTZONE MOISTURE AND SALT CONTENT IN REGIONS WITH SHALLOW GROUNDWATER". Your latest evaluation of the manuscript was as follows:

> "While the reviewers are generally satisfied, the first Reviewer raises a few (very) minor technical issues (mostly related to grammar) and one technical point related to provide a clear distinction between surface irrigation (flooding) and surface & ground water sources used to supply irrigation water. I would encourage the Authors to take all of the points into account. Once this is accomplished, I will then be in a position to make a final decision."

Below we have replied to the comments of reviewer point by point. In our response and in the revised manuscript we show in blue the changed text.

We are grateful to you and the reviewers for the comments and your time. We are looking forward to hearing from you whether additional changes are needed.

With high regard

Zalin Huo, Tammo Steenhuis and Zhongyi Liu

**Responses to the comments of Reviewer #2:**

**Page2. Number1**: In light of the many uses for the word performance, perhaps "optimum management of irrigated crops" would be more straightforward.

**Response:** Thank you for your suggestion. In the revised manuscript, the sentence was revised as "Optimum management of irrigated crops in regions with shallow saline groundwater requires a careful balance between application of irrigation water and upward movement of salinity from the groundwater." in line 23-25.

**Page3. Number1**: Page 3: Actual evporation

**Response:** Apologizes for the spelling mistake. In the revised manuscript, it was revised as "Actual evaporation".

**Page 4. Number1**: This is repetitive of the previous sentence.

**Response:** Thank you for your suggestion and this sentence was deleted in the revised manuscript.

**Page 4. Number 2**: In my previous comments, I mentioned that it would help some readers if surface irrigation was distinguished from irrigation supplied from surface water sources (as opposed to groundwater sources). Although the authors have clarified that this research was conducted in flood irrigated fields, I think this is still pertinent in this introduction to differentiate the irrigation methods from the source water when using the term "surface irrigation". This is especially relevant here because saline groundwater complicate the management salinity, regardless of irrigation technique (flood/surface, sprinkler, sub, or drip).

**Response:** We are grateful for your suggestion. In the revised manuscript, it was revised as "In arid and semi-arid areas where people divert surface water for flood irrigation and have poor drainage infrastructures, the groundwater table is close to the surface because more water has been applied than crop evapotranspiration" in line 53-55.

**Page 4. Number 3:** format problem

**Response:** Apologizes for the format mistake. Please see our response to the comment of Page 4, Number 2 above.

**Page 4. Number 4:** deeding

**Response:** Apologizes for the spelling mistake. It should be "seeding" and in the revised manuscript, the sentence was revised as "In north China, the fields are commonly irrigated in the autumn before soil freezing to leach salts and provide water for first growth after seeding in the following year (Feng et al., 2005)" in line 63-65.

**Page 6. Number 1: "**Detailed spatially**"

**Response:** Thanks for your reminder and the sentence was revised as "Detailed spatial input of soil hydrological properties and crop growth are required to take advantage of the model complexity (Flint et al., 2002; Rosa et al., 2012)" in line 95-97 in the revised manuscript.

**Page 8. Number 1:** "staring"

**Response:** Apologizes for the spelling mistake. In the revised manuscript, the sentence was revised as "This is followed by detailed description of the two-year field experiments started in 2017 in the Hetao irrigation district where maize and sunflower were irrigated by flooding the field" in line 144-146.

**Page 9. Number 1:** "The model was a proof of concept with calibrated values for evapotranspiration and soil salinity and was not simulated."

**Response:** In the revised manuscript, the sentence was revised as "The model was a proof of concept with calibrated values for evapotranspiration and soil salinity which was not simulated" in line 156-157.

**Page 9. Number 2:** This acronym is applied inconsistently throughout the text. See lines 219-220. Since your model acronym (EPICS) is very similar, I suggest that you ensure that this is correct and consistent to help the reader keep it all straight.

**Response:** Apologizes for the inconsistent acronym. For consistency of this acronym in the paper, we revised it as "Erosion Productivity Impact Calculator". The sentence was revised as "The new model that combines parts of the EPIC (Erosion Productivity Impact Calculator, Williams et al., 1989) with Shallow Vadose Groundwater model is called the *Evaluation of the Performance of Irrigated Crops and Soils* (EPICS)" in line 160-163.

**Page 12. Number 1:** This acronym is defined inconsistently here and above. This is the correct attribution from Williams et al., 1989. See line 162.

**Response:** Please see our response to the comment of Page 9, Number 2 above.

**Page 15. Number 1:** For modeling the daily soil moisture content and groundwater depth, first we need calculate the soil moisture content at field capacity and the drainable porosity based on the soil moisture characteristic curve.

**Response:** In the revised manuscript, the sentence was revised as "For modeling the daily soil moisture content and groundwater depth, first we need to calculate the soil moisture content at field capacity and the drainable porosity based on the soil moisture characteristic curve" in line 279-281.

**Page 16. Number 1:** Grammatical errors: Besides, considering the water and salt movement is different when there have irrigation and/or precipitation, we simulate the daily soil moisture content and salt with downward flux or upward flux.

**Response:** We are grateful for your reminder. In the revised manuscript, the sentence was revised as "Besides, we assume that the water and salt moves downward on rainy and/or irrigation days, while the water and salt moves upward on days without rain and/or irrigation**"** in line 281-283.

**Page 16. Number 2:** Lines 271-275: Both of these sentences have multiple grammatical errors, and should be reviewed to ensure that the authors' intented meaning is correctly stated.

**Response:** Apologizes for the grammatical errors. Please see our responses to the comment of Page 15, Number 1 and Page 16, Number 1 above.

**Page 18. Number 1:** This sentence has multiple grammatical errors.

**Response:** Thanks for your reminder. The sentence was revised, as "During the downward flux period, the upward water flux from groundwater is zero. Under this condition, the model can output the daily soil moisture content of different soil layers, the percolation from each soil layer to the soil layer beneath, the discharge from soil water to groundwater, the salt concentration of groundwater and of soil water in each soil layer, and the groundwater depth" in line 340-344.

**Page 21. Number 1:** As in the section above, please fix grammatical errors in this sentence.

**Response:** Thanks for your reminder. It was revised as "Under this condition, the model can output the daily soil moisture content of different soil layers, the upward groundwater flux, the groundwater depth, and the salt concentration of groundwater and of soil water in each soil layer." in line 395-397.

**Page 23. Number 1:** Does salinity of this source water change seasonally?

**Response:** We measured the salinity of the irrigation water that diverted from the Yellow River three times during crop growth period and the change was small. The mean salinity of the irrigation water is only around 100 mg/L. In the revised manuscript, it was revised as "… Irrigation water originates from the Yellow River. The change of the irrigation water salinity is small and can be ignored during the crop growth period. The area has an arid continental climate…" in line 449-450.

**Page 25. Number 1:** Was the salinity of the irrigation source water measured? Was the actual salinity of the irrigation water used in the mass balance (equation 18 of the model)?

**Response:** Yes, the salinity of the irrigation source water was measured three times during crop growth period and the change of the salinity was small. The mean measured salinity of the irrigation water was used in the mass balance. And we

assumed it is unchanged during the crop growth period. In the revised manuscript, it was revised as "… The fields were irrigated by flooding the field ranging from two to five times during the growing season (Table 1). The salinity of the irrigation source water was measured three times during crop growth period and the mean value was used in the mass balance. The salinity of the irrigation source water is assumed unchanged" in line 472-474.

**Page 32. Number 1:** As noted in the previous revision, it is difficult to distinguish between the blue dot markers that are in the legend from those that represent data. In five of these panels above (Figure 5), the legend lies within the data range. Possibly it would be better to use only one legend for all six panels and have it located outside the domain of the data. The same problem is observable in figure 6.

**Response:** Thank you for your suggestion. The legend and figures were revised as follows:

[Figure]

Fig. 5 Observed (blue dots) and simulated soil moisture content of the Shahaoqu experimental fields during model calibration (a,b,c) and validation (d,e,f)

[Figure]

Fig. 6 Observed (blue dots) and simulated soil salinity concentration of the experimental fields in Shahaoqu during model calibration (a,b,c) and validation (d,e,f).

**Page 49. Number 1:** Although dilution of salinity during irrigation events seems evident in the observed data, I would still recommend adding that future refinement of the model would be served by measuring the salinity of irrigation source water. This would be more important if this model was implemented for irrigation that depends on groundwater sources, especially hydrologically closed basins.

**Response:** Thank you for your suggestion. As we explained before, the mean salinity of the irrigation water from the Yellow River is around 100 mg/L according to the measured data and the change of the salinity can be ignored during the crop growth period. Thus we didn't consider the change of salinity of the irrigation water in this study. However, as the reviewer commented, the salinity of irrigation source water would be more important if this model was implemented for irrigation that depends on groundwater sources, especially hydrological closed basins. Therefore, we add this in the revised manuscript as follows:

"To obtain more accurate results in the future, the upward capillary flux from

groundwater needs to be improved. Also, future refinement of the model would be served by measuring the salinity of irrigation source water. This would be more important if this model was implemented for irrigation that depends on groundwater sources, especially hydrologically closed basins. In addition…" in line 862-865.

---

## Author Response (AR3)

**Revision Notes (HESS2019656)**

July 17, 2020

Dear Professor Alberto Guadagnini,

Thank you for allowing us to resubmit the minor changed manuscript Hess-2019-656 entitled "A FIELD VALIDATED SURROGATE CROP MODEL FOR PREDICTING ROOTZONE MOISTURE AND SALT CONTENT IN REGIONS WITH SHALLOW GROUNDWATER". Your latest evaluation of the manuscript was as follows:

> "While I do think the manuscript can be accepted at this stage, there are a few technical elements, related to the use of English, which should be carefully considered. For example, statements such as "Also, future refinement of the model would be served by measuring the salinity of irrigation source water. This would be more important if this model was implemented for irrigation that depends on groundwater sources, especially hydrologically closed basins" are somehow awkward in terms of wording. I am confident these issues will be resolved in this final segment of the process."

We have reviewed the full text and made some revisions. In our response and in the revised manuscript we show in blue the changed text. We are grateful to you and the reviewers for the comments and your time. We are looking forward to hearing from you whether additional changes are needed.

With high regard

Zalin Huo, Tammo Steenhuis and Zhongyi Liu

**List of all relevant changes made in the manuscript:**

[revised manuscript text omitted]

**Key words:** Surrogate hydrological model, irrigated crops, shallow aquifer

Nomenclature

[revised manuscript text omitted]